# Machine learning sequence prioritization for cell type-specific enhancer design

Alyssa J Lawler[1,2,3]*†, Easwaran Ramamurthy[1,3], Ashley R Brown[1,3], Naomi Shin[1,3], Yeonju Kim[1,3], Noelle Toong[1,3], Irene M Kaplow[1,3], Morgan Wirthlin[1,3], Xiaoyu Zhang[1,3], BaDoi N Phan[1,3,4], Grant A Fox[1,3], Kirsten Wade[5], Jing He[6,7], Bilge Esin Ozturk[8], Leah C Byrne[6,8,9,10], William R Stauffer[6], Kenneth N Fish[5], Andreas R Pfenning[1,3]*

[1]Computational Biology Department, School of Computer Science, Carnegie Mellon University, Pittsburgh, United States; [2]Biological Sciences Department, Mellon College of Science, Carnegie Mellon University, Pittsburgh, United States; [3]Neuroscience Institute, Carnegie Mellon University, Pittsburgh, United States; [4]Medical Scientist Training Program, University of Pittsburgh, Pittsburgh, United States; [5]Department of Psychiatry, Translational Neuroscience Program, University of Pittsburgh, Pittsburgh, United States; [6]Department of Neurobiology, University of Pittsburgh, Pittsburgh, United States; [7]Systems Neuroscience Center, Brain Institute, Center for Neuroscience, Center for the Neural Basis of Cognition, Pittsburgh, United States; [8]Department of Ophthalmology, University of Pittsburgh, Pittsburgh, United States; [9]Division of Experimental Retinal Therapies, Department of Clinical Sciences & Advanced Medicine, School of Veterinary Medicine, University of Pennsylvania, Philadelphia, United States; [10]Department of Bioengineering, University of Pittsburgh, Pittsburgh, United States

*For correspondence:
alawler@broadinstitute.org
(AJL);
apfenning@cmu.edu (ARP)

Present address: †Broad Institute of Harvard and MIT, Stanley Center for Psychiatric Research, Cambridge, United States

**Abstract** Recent discoveries of extreme cellular diversity in the brain warrant rapid development of technologies to access specific cell populations within heterogeneous tissue. Available approaches for engineering-targeted technologies for new neuron subtypes are low yield, involving intensive transgenic strain or virus screening. Here, we present Specific Nuclear-Anchored Independent Labeling (SNAIL), an improved virus-based strategy for cell labeling and nuclear isolation from heterogeneous tissue. SNAIL works by leveraging machine learning and other computational approaches to identify DNA sequence features that confer cell type-specific gene activation and then make a probe that drives an affinity purification-compatible reporter gene. As a proof of concept, we designed and validated two novel SNAIL probes that target parvalbumin-expressing (PV+) neurons. Nuclear isolation using SNAIL in wild-type mice is sufficient to capture characteristic open chromatin features of PV+ neurons in the cortex, striatum, and external globus pallidus. The SNAIL framework also has high utility for multispecies cell probe engineering; expression from a mouse PV+ SNAIL enhancer sequence was enriched in PV+ neurons of the macaque cortex. Expansion of this technology has broad applications in cell type-specific observation, manipulation, and therapeutics across species and disease models.

## Editor's evaluation

This article describes an exciting new approach for tagging and isolation of unique neuronal subpopulations based on machine learning selection of cell-specific enhancer elements in the genome. The article highlights a specific test case of this technology with neurons expressing Parvalbumin, but this method could be applied to any neuronal or even non-neuronal cell type. The tools

and overall approach described here will enable cell tagging in model organisms for which transgenic lines are not commonly available or even expression of other transgenes for control of cell function or genetic perturbation.

## Introduction

The biology of the brain is complicated by vast diversity in cell types, subtypes, and states. Contemporary advancements in single-cell sequencing have identified over a hundred molecularly distinct neuronal populations in the mammalian cortex (*Lake et al., 2016*; *Saunders et al., 2018*; *Tasic et al., 2018*; *Hodge et al., 2019*; *Zeisel et al., 2015*), including several small subpopulations of gamma aminobutyric acid (GABA)ergic neurons with specialized functions that are critical for the control of neuronal inhibition (*Kepecs and Fishell, 2014*; *Lim et al., 2018*). Understanding neurological function in health and disease from a cell type-specific perspective is critical to the progress of neuroscience.

Such endeavors necessitate cell type-specific technologies to identify, isolate, and manipulate discrete cell populations. Transgenic mouse strains targeting major inhibitory neuron subclasses, including parvalbumin-expressing (PV+), somatostatin-expressing (SST+), and serotonergic (5-HT+) neurons, are widely used today and have been instrumental toward our understanding of these cell types (*Madisen et al., 2010*; *Taniguchi et al., 2011*). More recently, many developers have turned toward scalable virus-based cell type-specific tools (*Dimidschstein et al., 2016*; *Hrvatin et al., 2019*; *Nair et al., 2020*; *Vormstein-Schneider et al., 2020*; *Graybuck et al., 2021*; *Mich et al., 2021*). Adeno-associated virus (AAV) technologies became particularly attractive with the invention of AAV variants that cross the blood–brain barrier to transduce the central nervous system, AAV-PHP.B, and AAV-PHP.eB (*Chan et al., 2017*; *Deverman et al., 2016*). In line with certain transgenic engineering, an emerging AAV targeting strategy is to incorporate cell type-specific enhancer elements into the viral genome to promote restricted expression. Enhancer activity can be extremely selective, even more so than the activity of most genes and their associated promoters (*Hoffman et al., 2013*; *Kellis et al., 2014*; *Roadmap Epigenomics Consortium et al., 2015* 2015). Thus, enhancers may be used to confer specificity for neuron subtypes that cannot be resolved by the expression of a single marker gene (*Tasic et al., 2018*) or for which the marker gene promoter is not specific on its own (*Nathanson et al., 2009*).

Despite the enthusiasm for enhancer sequences in cell type-specific AAV development, their selection remains challenging. ATAC-seq (*Buenrostro et al., 2013*) is an advantageous technique for defining potential cell type-specific enhancer regions because of its high nucleotide resolution and its compatibility with small cell populations and even with single-cell technologies (*Buenrostro et al., 2015b*; *Cusanovich et al., 2015*). Though convenient, chromatin accessibility is a noisy approximation for enhancer activity and differentially accessible sequence elements often fail to produce selective expression in vivo. A parallel screening approach involving single-nucleus sequencing of barcoded enhancer libraries, PESCA, was proposed to speed up the selection process toward a successful enhancer-driven virus (*Hrvatin et al., 2019*). Other approaches leveraged marker gene proximity and conservation across species for enhancer prioritization (*Vormstein-Schneider et al., 2020*). These efforts highlight the low conversion rate from experimentally suggested cell type-specific open chromatin regions (OCRs) to desired cell type-specific activity. We hypothesized that machine learning models could be applied as additional in silico filters to reduce the burden of experimental screening in cell type-specific AAV development while maintaining the convenience of ATAC-seq as the sole input data requirement.

The nucleotide sequence code that links transcription factor (TF) binding sites and other DNA features to enhancer activity is underutilized in AAV enhancer design, perhaps due to its complexity (*Jindal and Farley, 2021*). We reasoned that machine learning classifiers could be leveraged to identify the most characteristic and specific enhancer sequence patterns for a cell population, enabling efficient prioritization of sequences that are likely to drive selective expression. Convolutional neural networks (CNNs) (*Le Cun et al., 1989*) and support vector machines (SVMs), for example, have achieved state-of-the-art performance on predicting enhancer activity from sequence (*Arvey et al., 2012*; *Ghandi et al., 2014*; *Zhou and Troyanskaya, 2015*). Here, we present a framework for machine learning-assisted engineering of cell type-specific AAVs, which we refer to as Specific Nuclear-Anchored Independent Labeling (SNAIL). Similar to our previously described Cre-activated

AAV technology cSNAIL (*Lawler et al., 2020*), SNAIL probes have the unique advantage of an affinity purification-compatible fluorescent reporter that can be used to isolate rare cell types (*Mo et al., 2015*; *Deal and Henikoff, 2010*; *Lawler et al., 2020*). Unlike cSNAIL, which relies on Cre activation, SNAIL probes are stand-alone AAVs driven by cell type-specific enhancer sequences selected through machine learning models. Thus, SNAIL probes may be used in a wide variety of systems, including wild-type mice, primates, and other species.

We used our framework to create two novel AAV probes for PV+ neurons. In the mouse primary motor cortex, PV+ SNAIL probes labeled PV+ neurons with >70% specificity when compared to Pvalb antibody staining. PV+ SNAIL probes in this context were more specific to GABAergic PV+ neurons than the common Pvalb-2A-Cre/Ai14 mouse strain. In addition, isolated populations of tagged cells from the cortex, striatum, and external globus pallidus (GPe) were each heavily enriched for known PV+ open chromatin signatures. Nucleotide-resolution model interpretation highlighted a collection of 14 TF binding motif families contributing to PV+ neuron-specific enhancer activation. Machine learning can aid in cross-species probe design as well; at least one SNAIL probe showed PV+ neuron selectivity in both mouse and macaque. These results demonstrate concrete utility in sequence-level information for AAV enhancer selection, setting the stage for efficient probe design for a wide range of cell types.

## Results

### SVMs discriminate known cell type-specific regulatory sequences

We built machine learning classifiers, primarily SVMs, to discriminate sequences of differential OCRs between two cell populations. These models take 500 bp candidate enhancer DNA sequence strings as input and they output, for each sequence, the cell type in which that candidate enhancer is active and an associated score. The similarity between sequences is determined based on gapped k-mer count vectors, that is, the number of occurrences of all short subsequences of length k, tolerating some gaps or mismatches, as implemented by LS-GKM (*Ghandi et al., 2014*; *Lee, 2016*). Where there are sufficient reliable sequence features associated with differential enhancer activation between two cell types, the model should learn these principles during the training phase and then be able to apply these principles to determine the cell type-specific activities of new sequences. We imposed upfront that training sequences have a minimum fold difference in chromatin accessibility between the two cell types used to define the classes. This should help the model learn cell type-specific features of enhancer activation rather than general enhancer features. We chose this strategy because it closely aligned with our goal of prioritizing sequences that would activate enhancers in one cell type and not others.

To evaluate this strategy, we first built SVMs comparing select broad classes of cell types in the brain using publicly available single-nucleus (sn)ATAC-seq data from the mouse motor cortex (Mop) (*Li et al., 2021*; *Figure 1—figure supplement 1*). These were (i) a neuron vs. astrocyte classifier and (ii) an excitatory neuron vs. inhibitory neuron classifier. To assess model performance, we used standard classifier metrics, the area under the receiver operator curve (auROC), and the area under the precision-recall curve (auPRC). These scores quantify model performance by comparing the predicted class to the actual class, where a randomly guessing binary classifier would have an auROC score of 0.5 and an auPRC score equal to the fraction of actual positives in the data, and a perfect classifier would have a maximum auROC score of 1.0 and auPRC score of 1.0. Both models performed well on held-out data, achieving auROCs of 0.95 and 0.93 and auPRCs of 0.98 and 0.82 (*Figure 1—figure supplement 2*).

Although the models were built for enhancer sequences, they recapitulated known cell type-specific activation patterns of some commonly used AAV promoter sequences: *Gfap*, *Camk2a*, and *Dlx* (*Figure 1—figure supplement 2*). The *Gfap* promoter sequence, which has a heavy astrocyte bias in vivo, was classified as astrocyte-specific in our neuron vs. astrocyte model (5580 astrocyte-specific enhancers evaluated; *Gfap* ranks 3298/5580). The same neuron vs. astrocyte model scored the *Camk2a* promoter and *Dlx* promoter sequences, which are known to have neuron-specific activity, as neuron-specific (14,347 neuron-specific enhancers evaluated; *Camk2a* = rank 13,300/14,347, Dlx = rank 9736/14,347). Also consistent with empirical expectations, the excitatory vs. inhibitory neuron model predicted the *Camk2a* sequence to have an excitatory neuron preference and the *Dlx* sequence to have an inhibitory neuron preference (15,391 excitatory neuron-specific enhancers evaluated; *Camk2a*

= rank 11,541; 4,608 inhibitory neuron-specific enhancers tested; Dlx = rank 1142/4608) (*Figure 1—figure supplement 2*). Therefore, this classification strategy is capable of correctly predicting cell type-specific regulatory sequence activity in the viral context for these very distinct cell classes.

## Machine learning models accurately predict PV+ neuron-specific open chromatin from sequence

Next, we assessed whether the same modeling strategy could be applied to more narrowly defined neuron subtypes, using PV+ neurons as a target. Again, we trained binary classifiers on sequences underlying differential ATAC-seq peaks between two cell populations (PV+ neurons and another population). To maximize sequence signal detection, we built several models comparing PV+ neurons to either PV- cells generally, excitatory neurons, VIP+ neurons, or SST+ neurons. In addition, we varied the input data source to be either 'population-derived' (from bulk ATAC-seq sequencing on purified nuclei populations) or 'sn-derived' (from single-nucleus open chromatin assays such as droplet-based snATAC-seq). Finally, we implemented multiple modeling strategies, including linear gapped k-mer SVMs, nonlinear gapped k-mer SVMs using the radial basis function (rbf), and CNNs. The ratios of positive and negative examples during training ranged from 1:0.4 to 1:3.7. Detailed information about all model parameters and training, validation, and test data is available in *Figure 1—source data 1* (SVMs) or *Figure 1—source data 2* (CNNs).

First, to define potential PV+ neuron and PV- cell enhancer sequences in the mouse cortex in a population data-driven manner, we conducted ATAC-seq on the PV+ and PV- nuclei of Pvalb-2A-Cre mice. We isolated the nuclei populations using previously described Cre-dependent AAV affinity purification technology, cSNAIL (*Lawler et al., 2020*). cSNAIL probes activate an isolatable nuclear envelope tag in the presence of Cre recombinase protein. Therefore, purified populations from these mice are a direct reflection of cells labeled by the Pvalb-2A-Cre mouse strain, a current standard for PV+ neuron labeling. Population-derived PV+ vs. PV- SVMs could predict the correct classification on held-out data with high accuracy (test set auROC = 0.87–0.88, auPRC = 0.95–0.96; gray lines *Figure 1b and c*), indicating that there were substantial sequence pattern differences between the PV+ and PV- classes and that the models were able to learn these differences.

We then considered the possibility that the PV+ vs. PV- models were learning features of general neuron vs. glia enhancer sequence properties and not necessarily features specific to PV+ neurons as the PV- data contained a high proportion of glial cells, a developmental outgroup to neurons. To address this issue, we trained additional population-derived SVMs that directly discriminated between enhancer sequences of PV+ neurons and other neuron subtypes using publicly available ATAC-seq data from INTACT-sorted excitatory (EXC) neurons and VIP+ neurons (*Mo et al., 2015*). These models performed well (auROC = 0.89–0.93, auPRC = 0.79–0.92; *Figure 1b and c*), demonstrating that, even at the level of neuron subtypes, OCR sequence information is rich enough to reliably distinguish cell type-specific activity.

To survey an additional machine learning strategy, we built CNN classifiers from the same underlying data (*Figure 1—figure supplement 3*). CNNs are a type of artificial neural network defined by multiple convolutional layers. The CNNs were trained to take in 1000 bp DNA sequence strings with different accessibility between two cell types, automatically extract predictive sequence features, and output a cell class probability between 0 and 1 (see Materials and methods for details). Compared with SVMs, CNNs are better equipped to learn higher-order interactions between sequence features due to CNNs' capacity for flexible feature representation and automated feature selection (*Le Cun et al., 1989*). The CNNs were highly accurate (auROC = 0.95–0.96, auPRC = 0.76–0.97; *Figure 1d*). Thus, CNNs represent an additional successful approach to discriminate OCR sequence differences between purified neuron populations.

While ATAC-seq from purified cell populations is advantageous for its depth and recovers many examples of differentially accessible reads between neuron subtypes, many neuron populations of interest are not yet isolatable, even through transgenic means. Single-nucleus sequencing technologies can be applied to measure neuron subtype-resolution open chromatin without cell sorting by performing several parallel microreactions that introduce unique cell barcodes into ATAC-seq sequencing reads and then clustering the cells with similar chromatin profiles. We explored whether cell type-specific enhancer sequences derived from mouse motor cortex snATAC-seq (*Li et al., 2021*) were sufficient to produce neuron subtype-level classifiers. We trained five pairwise linear gapped

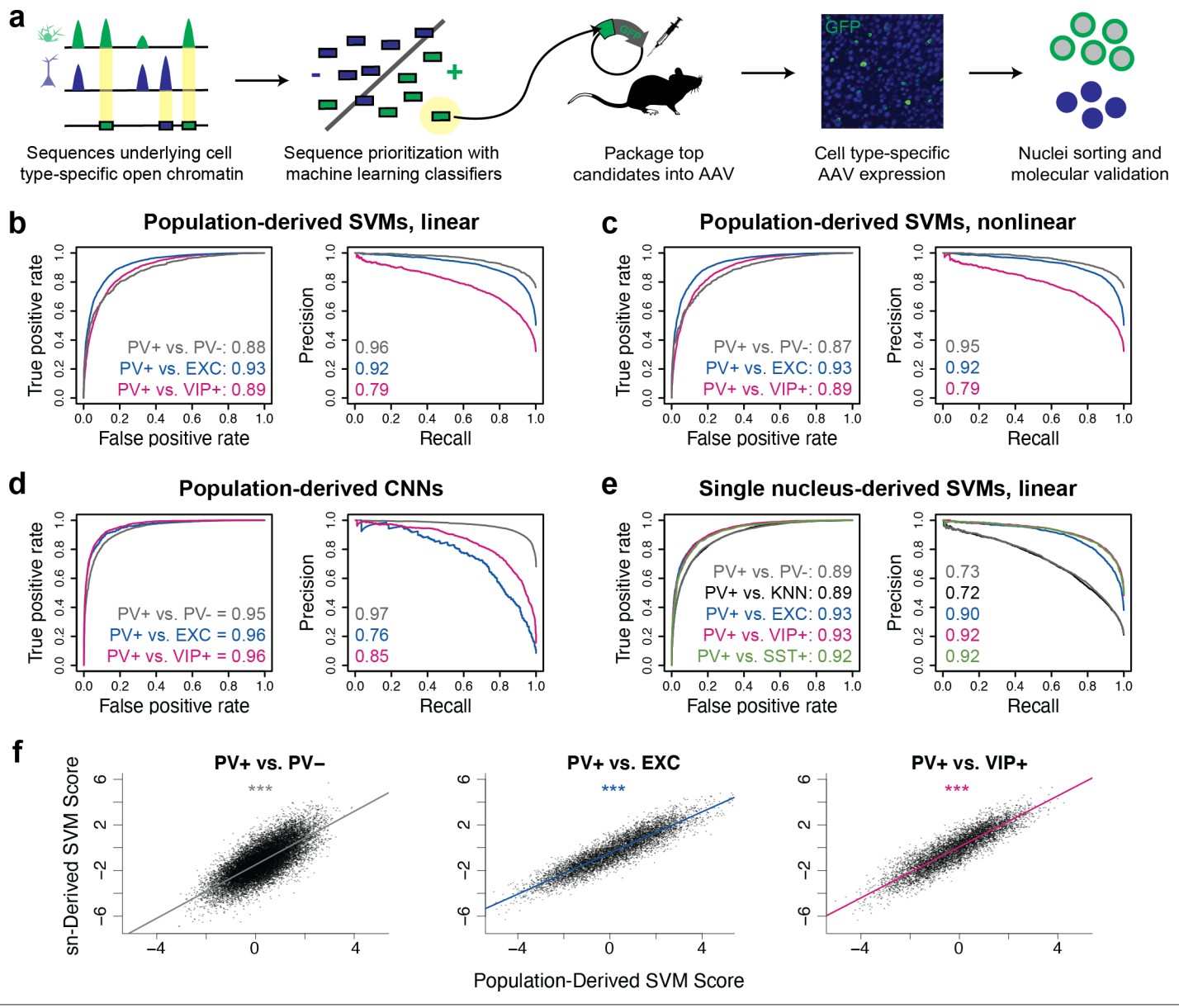

**Figure 1.** Classification of neuron subtype-specific enhancer activity from sequence. (**a**) Schematic representation of the Specific Nuclear-Anchored Independent Labeling (SNAIL) workflow. (**b–e**) Receiver operator characteristic and precision-recall performance metrics for various cell type-specific enhancer sequence model strategies and data modalities. The reported numbers are the areas under the curves for each model. (**f**) Scatter plots for support vector machine (SVM) scores reported by equivalent population-derived models and single-nucleus-derived models. ***p-Value of Pearson correlation <0.001.

The online version of this article includes the following source data and figure supplement(s) for figure 1:

**Source data 1.** Support vector machine (SVM) parameter tuning and performance evaluations.

**Source data 2.** Convolutional neural network (CNN) parameter tuning and performance evaluations.

**Source data 3.** Model scores on externally screened parvalbumin-expressing (PV+) enhancer candidates.

**Source data 4.** Identification of motif sites within external parvalbumin-expressing (PV+) neuron enhancer screen, E1–E34.

**Figure supplement 1.** snATAC-seq cluster assignments.

**Figure supplement 2.** Validation of modeling strategy using known promoters targeting broad cell classes.

**Figure supplement 3.** Convolutional neural network (CNN) strategy overview.

**Figure supplement 4.** Pearson correlations of enhancer scores between support vector machines (SVMs).

*Figure 1 continued on next page*

*Figure 1 continued*

**Figure supplement 5.** Comparison to alternative enhancer prioritization strategies.

**Figure supplement 6.** Interpretation of top external parvalbumin-expressing (PV+) adeno-associated virus (AAV) enhancer sequences.

k-mer SVMs to discriminate differential open chromatin sequences from snATAC-seq clusters or groups of clusters. These included analogous models to the population-derived models comparing PV+ vs. PV-, PV+ vs. EXC, and PV+ vs. VIP+. In this case, the single-nucleus-derived PV+ vs. PV- model refers to a model trained on differential OCR sequences comparing PV+ cluster nuclei to all other nuclei with a random sampling probability. The PV+ vs. k-nearest-neighbor (KNN) model is an additional variation on the PV+ vs. PV- model where the PV- nuclei sampling for differential OCR analysis was selected for similarity to the PV+ cluster as implemented in SnapATAC (*Fang et al., 2021*). We also produced a model comparing PV+ vs. SST+ neurons, the most similar subtype to PV+. Single-nucleus-derived SVMs were also able to classify cell type-specific enhancer sequences with high accuracy (auROC = 0.89–0.93, auPRC = 0.72–0.92; *Figure 1e*).

Moreover, models built independently from different data sources identified similar sequence contributions for equivalent tasks. When scoring the population-derived sequences through both the population-derived SVMs and the single-nucleus-derived SVMs, individual sequences had highly similar scores (*Figure 1f*). These findings highlight the prevalence of reliable cell type-specific enhancer sequence signatures that can be defined by a variety of classifier types and sources of open chromatin measurements. The parameter and performance details of all models can be found in *Figure 1— source data 1* (SVMs) and *Figure 1—source data 2* (CNNs).

## Models learn biological signatures relevant for AAV probe design

To ensure that the neuron subtype-level models were identifying signatures that were relevant for the specific purpose of creating selective PV+ neuron viruses, we evaluated model predictions on 34 externally validated successful and unsuccessful PV+ probe enhancer candidates from *Vormstein-Schneider et al., 2020*, named E1–E34. Importantly, the enhancer sequence from the probe with the lowest PV+ specificity (E4; 14% specificity) received a negative score from every model, and two of the enhancer sequences with highest cortical PV+ specificity (E22 and E29; 94% specificity) received high positive scores from every model (*Figure 1—source data 3*).

We binarized PV+ enhancer AAV sequences E1–E34 into either the high-specificity (>70%) or low-specificity (<70%) group and assessed the relationships between these groups and ATAC-seq log2 fold difference or SVM score (*Figure 1—figure supplement 5*). These groups had little variation in mammalian nucleotide sequence conservation levels measured by PhyloP, but many of the sequences with highly confidently predicted PV+-specific accessibility had PV+ accessibility in humans (*Figure 1—figure supplement 5b*). On the complete set of enhancers, SVM score was predictive of the PV+ specificity group (p=0.008), and differential activity log2 fold difference had a weak association with PV+ specificity group (p=0.069) (*Figure 1—figure supplement 5c*). Much of the log2 fold difference association with group PV+ specificity was driven by enhancer sequences with low log2 fold difference and low PV+ specificity. Undesirably, there were some sequences with high log2 fold difference and low in vivo specificity. Within the subset of enhancers with high log2 fold difference (log2 fold difference > 1), the log2 fold difference was not associated with specificity group (p=0.601), while SVM score was weakly associated (p=0.057) (*Figure 1—figure supplement 5d*). Unlike log2 fold difference scores alone, SVM scores may limit false-positive candidates and improve efficient enhancer AAV selection.

This result emphasizes the benefit of enhancer pre-selection with machine learning, which could reduce in vivo screening efforts by identifying the best PV+ enhancer sequences before experimentation. The models predicted which PV+ enhancer sequence candidates were likely to be cell type-specific expression drivers. They also carry information about precisely which subsequences were responsible for the predicted PV+ neuron-specific activation. Sequence E29, within the *Inpp5j* locus, was predicted to have PV+ neuron-specific activity due to a central Mef2 motif site and nearby Esrrg motif site, among others (*Figure 1—figure supplement 6*, *Figure 1—source data 4*). Sequence E22, within the *Tmem132c* locus, was predicted to have PV+ specificity in part due to Nkx28 and Lhx6 motif sites (*Figure 1—figure supplement 6*, *Figure 1—source data 4*). Nevertheless, none of these

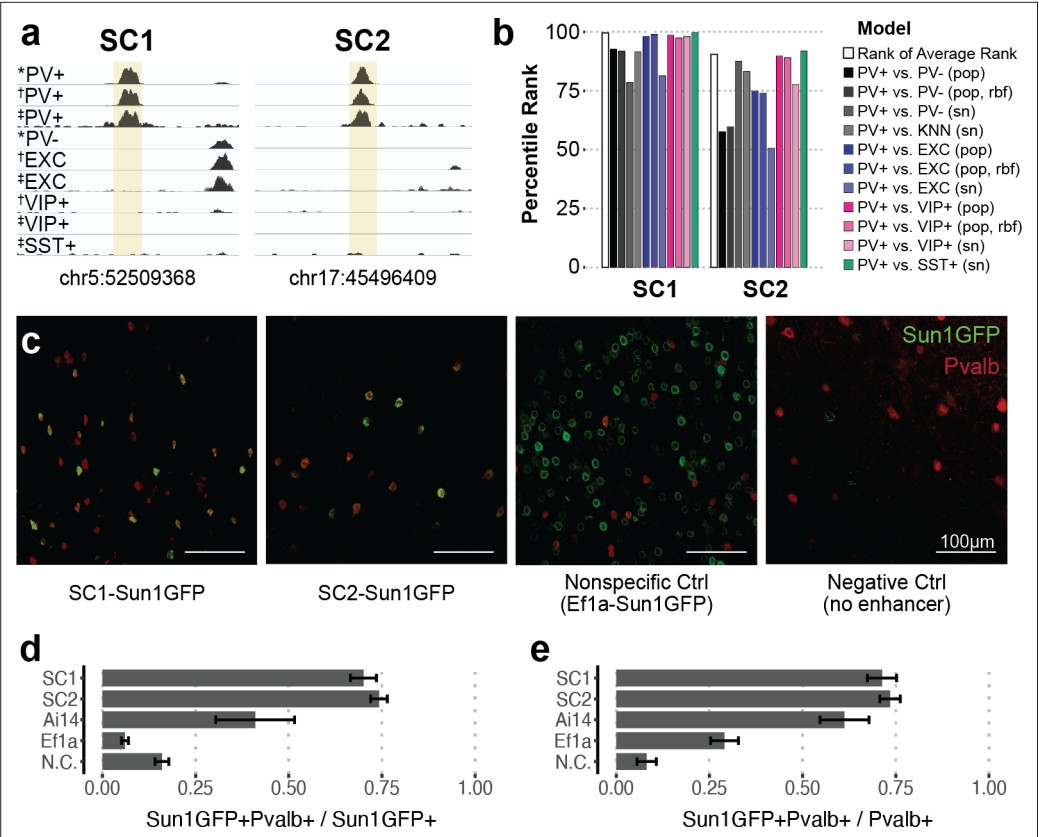

**Figure 2.** Two sequence candidates selectively activate adeno-associated virus (AAV) expression in parvalbumin-expressing (PV+) neurons. (**a**) Genome browser visualization of PV+-specific ATAC-seq signal at sequence candidates SC1 and SC2. * cSNAIL data, † INTACT data from *Mo et al., 2015*, ‡ snATAC-seq from *Li et al., 2021*. (**b**) Percentile rank of support vector machine (SVM) scores among 1755 true PV+-specific enhancer sequence candidates that scored positively across all models. Linear population-derived models are denoted with 'pop,' nonlinear population-derived models are denoted with 'pop, rbf,' and linear single-nucleus-derived models are denoted with 'sn.' (**c**) Example images of AAV Sun1GFP expression against parvalbumin (Pvalb) antibody staining. (**d, e**) Quantification of AAV Sun1GFP or Pvalb-2A-Cre/Ai14 reporter overlap with Pvalb+ cells. Bar heights represent the mean among images, and the error of the mean is shown. N cells = 1322 (SC1), 2570 (SC2), 1340 (Ai14), 2013 (*Ef1a*), and 504 (N.C.). N.C., negative control.

The online version of this article includes the following source data for figure 2:

**Source data 1.** Detailed information for 1755 experimental parvalbumin (PV)-enriched open chromatin regions (OCRs) with positive parvalbumin-expressing (PV+) model scores.

**Source data 2.** Image quantification for Specific Nuclear-Anchored Independent Labeling (SNAIL) viruses or the Pvalb-2A-Cre mouse strain reporter with Pvalb immunohistochemistry.

---

enhancers were our highest predicted PV+ neuron sequences, so we continued to investigate additional enhancer candidates genome-wide for PV+ SNAIL probe implementation.

## Two candidate PV+ SNAIL probes successfully target PV+ neurons in the mouse cortex

Based on the predictions of all PV+ enhancer models on our candidates, we prioritized two highly characteristic PV+ neuron enhancer sequences and tested their ability to drive targeted expression in vivo (*Figure 2*). We refer to these sequence candidates as SC1 and SC2. Among true PV+ neuron-specific enhancer sequences that (i) were differential OCRs in PV+ vs. PV-, PV+ vs. EXC, and PV+ vs. VIP+ sorted population data and (ii) scored PV+ positive across all SVM evaluations (1755 sequences), SC1 was the highest predicted sequence candidate, while SC2 was in the 90th percentile (*Figure 2b*, *Figure 2—source data 1*). We chose these sequences to represent two different confidence levels

and evaluate the general potential of the top 10% of machine learning-prioritized PV+ neuron-specific enhancers.

SC1 and SC2 sequences were cloned into separate vectors upstream of the cSNAIL reporter gene, *Sun1GFP*. To minimize off-target effects, PV+ SNAIL probes directly rely on transcriptional activation from SC1 or SC2, without a minimal promoter (see Materials and methods). We also prepared two control vectors: a negative control that was the identical vector but with no enhancer or promoter and a nonspecific control that was the identical vector but with a common *Ef1a* promoter sequence in place of the candidate sequence. When packaged with AAV-PHP.eB and delivered to the mouse through systemic injection, the SC1-Sun1GFP and SC2-Sun1GFP constructs promoted cortical fluorescence that was restricted to PV+ neurons, while the *Ef1a* virus did not (*Figure 2c–e*, *Figure 2—source data 2*). We quantified images with consistent positioning in the primary motor cortex, with representation from all layers. Compared with immunohistochemistry-labeled Pvalb protein, SC1 and SC2-mediated expression of Sun1GFP was restricted to Pvalb+ neurons in ~70–74% of cases. This was an 11-fold enrichment in precision over the *Ef1a* promoter and notably an almost twofold enrichment over Cre reporter labeling in Pvalb-2A-Cre/Ai14 double transgenic mice (*Figure 2c–e*, *Figure 2—source data 2*). We expect these to be conservative estimates of PV+ targeting due to incomplete antibody capture. On average, Sun1GFP expression from SC1 and SC2 SNAIL probes labeled ~71–73% of Pvalb+ neurons. The rate is limited by the transduction properties of the AAV-PHP.eB capsid, which is expected to transduce 55–70% of neurons in the cortex (*Chan et al., 2017*). The negative control virus was injected at as high of a concentration as possible (14× the concentration of SC1, SC2, and *Ef1a* injections) to detect any biases in spurious background expression. At this titer, a number of cells exhibited GFP expression (*Figure 2—source data 2*), but these did not appear biased toward PV+ neurons (*Figure 2d*).

## Isolation of PV+ SNAIL-labeled nuclei captures PV+ cortical interneurons

Expression of Sun1GFP differentiates SNAIL probes from other cell type-specific AAV technology. The stable nuclear envelope association of this tag enables affinity purification using magnetic beads coated with anti-GFP antibodies, which is advantageous for rare population isolation and downstream epigenetic assays. In many contexts, purification of a cell population is more efficient than single-nucleus sequencing technologies, especially if the population of interest is in low proportion or the desired downstream applications are not available in single-nucleus approaches. Taking advantage of this property, we isolated Sun1GFP-expressing nuclei induced by SC1-Sun1GFP, SC2-Sun1GFP, or *Ef1a*-Sun1GFP SNAIL virus from the mouse cortex and performed ATAC-seq. Through comparison with known PV+ neuron ATAC-seq (via cSNAIL in the Pvalb-2A-Cre strain) and PV- or bulk ATAC-seq, including cSNAIL PV- cell fractions and *Ef1a* virus signatures, we determined that both SC1-Sun1GFP and SC2-Sun1GFP cells were highly enriched for PV+ neurons.

The first principal component, accounting for 84% of the total variance, separated known PV+ neuron samples from PV- and bulk tissue samples. Likewise, SC1-Sun1GFP and SC2-Sun1GFP samples grouped with the PV+ samples while *Ef1a*-Sun1GFP samples grouped with the PV- and bulk sample signatures (*Figure 3a*). At the *Pvalb* locus, there were highly reproducible OCR signals between PV+ cSNAIL, PV+ snATAC-seq, SC1-Sun1GFP, and SC2-Sun1GFP samples that did not appear in bulk tissue, PV-, or *Ef1a*-Sun1GFP samples (*Figure 3b*).

A major goal for PV+ SNAIL probes was to replace transgenic mouse strain technologies in certain contexts. Ideally then, ATAC-seq from Sun1GFP-sorted cells from SNAIL probes in wild-type mice should provide similar information as ATAC-seq from Sun1GFP-sorted cells from cSNAIL in Pvalb-2A-Cre transgenic mice. Therefore, we defined PV+ cSNAIL ATAC-seq log2FoldDifference over bulk cortical tissue ATAC-seq as a gold standard for each OCR. For SC1 and SC2, we computed the correlations between the log2FoldDifference of OCR signal relative to bulk tissue and the log2FoldDifference of OCR signal in PV+ cSNAIL relative to bulk tissue. To establish an upper limit for correlation, we compared two different batches of cortical PV+ cSNAIL samples, which had a Pearson correlation of 0.86 and a Spearman correlation of 0.85. As a lower limit, we evaluated the nonspecific *Ef1a* control virus, which had a Pearson correlation of 0.38 and a Spearman correlation of 0.26. Because the AAV-PHP.eB capsid has a neuron bias, these lowly correlated signatures are likely to be general neuron specifications shared among PV+ and other neurons. Within this range, SC1 and SC2 had very high

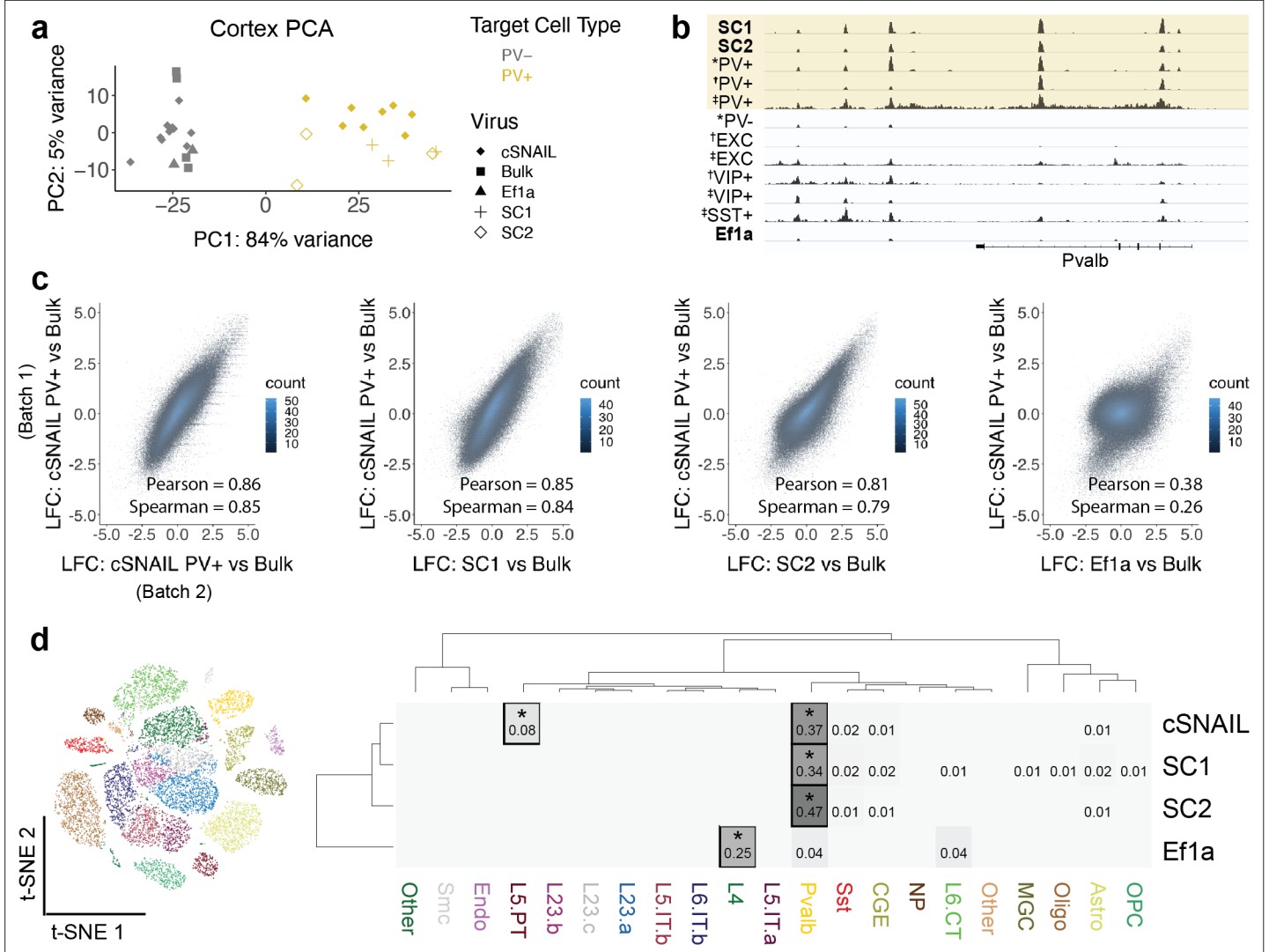

**Figure 3.** Cortical SC1 and SC2 Specific Nuclear-Anchored Independent Labeling (SNAIL)-isolated nuclei recapitulate parvalbumin-expressing (PV+) GABAergic interneuron ATAC-seq signatures. (**a**) Principal component analysis (PCA) of ATAC-seq counts across samples. (**b**) Genome browser visualization of ATAC-seq signal at the *Pvalb* gene locus. Tracks represent the pooled sample p-value signal. Each track of similar data type is normalized to the same scale: SNAIL data range 0–335, *cSNAIL data range 0–93, †INTACT data range 0–200, ‡snATAC-seq data range 0–2. (**c**) Scatter plots of ATAC-seq log2 fold difference relative to bulk tissue ATAC-seq, comparing PV+ cSNAIL to other adeno-associated viruses (AAVs). The density of overlapping points is shown by the plot color. (**d**) snATAC-seq nuclei clusters as visualized by t-SNE. The dendrograms show hierarchical clustering of Euclidean sample distances by Ward's minimum variance method D2. The heatmap shows the percentage of population open chromatin regions (OCRs) enriched relative to bulk that are also cluster-specific marker OCRs. *Hypergeometric enrichment p<0.01.

The online version of this article includes the following source data for figure 3:

**Source data 1.** Differential open chromatin region (OCR) statistics for parvalbumin-expressing (PV+) Specific Nuclear-Anchored Independent Labeling (SNAIL)-isolated cortical ATAC-seq relative to bulk tissue cortical ATAC-seq.

correlation with cSNAIL, with SC1 achieving almost equivalent correlation as the two cSNAIL batches (SC1 Pearson = 0.85 and Spearman = 0.84; SC2 Pearson = 0.81 and Spearman = 0.79) (***Figure 3c***). The details for differential OCRs in each virus relative to bulk tissue can be found in ***Figure 3—source data 1***.

Finally, we compared SC1-Sun1GFP+ and SC2-Sun1GFP+ cell open chromatin signatures to those of snATAC-seq clusters from the mouse motor cortex (***Figure 3d***; ***Li et al., 2021***). We defined cluster-specific OCRs for each snATAC-seq cluster and population-enriched OCRs for SNAIL-isolated cells relative to bulk tissue (see Materials and methods) and assessed the overlaps. We found that cSNAIL-isolated PV+ OCRs, SC1-isolated OCRs, and SC2-isolated OCRs were each significantly enriched for

PV+ cluster-specific markers (34–47% overlap, hypergeometric p=0), while OCRs from *Ef1a*-isolated cells were not enriched for PV+ cluster-specific markers (4% overlap, p=1). *Ef1a* OCRs instead had the highest enrichment for markers of a layer 4 excitatory neuron cluster (25% overlap, p=5.3 × 10$^{-5}$). We also note that cSNAIL PV+ ATAC-seq had an additional 8% overlap with excitatory cluster L5 PT markers (p=2.5 × 10$^{-45}$), possibly reflective of Pvalb-2A-Cre line labeling in layer 5 parvalbumin-expressing excitatory neurons (*Tanahira et al., 2009*; *Jinno and Kosaka, 2004*; *Roccaro-Waldmeyer et al., 2018*). These OCRs were absent in SC1- and SC2-isolated cells. In fact, SC1 and SC2 had no enrichment for cluster-specific OCRs of any cluster other than PV+ (≤2% overlap, p>0.1), including the closely related SST+ neuron population. This suggests that SC1 and SC2 SNAIL probes target a stricter subset of the cells than the Pvalb-2A-Cre mouse strain, likely restricted to PV+inhibitory interneurons.

## Chromatin accessibility differences between PV+ neurons in different brain regions

SC1 and SC2 SNAIL probes were designed based on the sequence properties of cortical PV+ neurons. Many PV+ neurons throughout the brain have a common developmental origin in the medial ganglionic eminence (MGE), but there are substantial OCR differences between mature PV+ neuron populations in different brain regions. From cSNAIL-isolated PV+ populations in Pvalb-2A-Cre mice (*Lawler et al., 2020*), we characterized thousands of OCRs with differential accessibility between the cortex, striatum, and GPe (p$_{adj}$<0.01, |log2FoldDifference| > 1) (*Figure 4a*, *Figure 4—source data 1*). These differences were associated with distinct TF binding motifs (*Figure 4b*, *Figure 4—source data 2*). For example, OCRs that were more accessible in cortical PV+ neurons relative to striatal and GPe PV+ had enrichment for Mef2 family motifs, which are important in neuroplasticity (*Donato et al., 2015*), may play a role in specifying or maintaining the MGE PV+ neuron lineage in mouse and human (*Mayer et al., 2018*), and have been linked to neurodevelopmental disorders involving the prefrontal cortex (*Mitchell et al., 2018*). TFs with motifs enriched in PV+ neuron OCRs that are more open in striatum relative to cortex and GPe included Tgif1, a key homeodomain gene involved in holoprosencephaly (*Taniguchi et al., 2012*). At 6654 differential OCRs, GPe-specific PV+ OCRs were the most unique and had TF motif enrichments, including the Lhx3, Pou5f1, Esrrg, and Pax3 motifs.

These molecular differences likely relate to functional differences, such as the tendency of PV+ cells in the GPe to project to other brain regions vs. the local nature of PV+ cells in the cortex (*Hernández et al., 2015*; *Saunders et al., 2016*). We used GREAT, which associates OCRs to one or more genes using proximity-based gene regulatory domains and then performs pathway enrichment analysis for those genes, to assess Gene Ontology enrichments in the brain region-specific PV+ ATAC-seq OCR sets relative to all PV+ OCRs (*Figure 4—source data 3*; *McLean et al., 2010*). The set of PV+ OCRs enriched in cortical PV+ neurons included 10 regions associated with the *Bdnf* gene (Ensembl Genes; false discovery rate [FDR] Q = 0.0035). *Bdnf* is generally expressed in excitatory forebrain neurons but not PV+ interneurons. The presence of PV+ neuron OCRs near *Bdnf* could represent genomic regions with nonenhancer functions, enhancers that regulate another gene, the binding regions of repressive TFs, or trace contamination from excitatory populations. Other cortex-specific PV+ enrichments included terms related to sensory perception, especially smell. Striatum-specific PV+ neuron OCRs were enriched for the adenylate cyclase-inhibiting dopamine receptor signaling pathway (GO:BP; FDR Q = 0.010) and bradykinesia (Mouse Phenotype; FDR Q = 0.046). OCRs preferentially open in GPe PV+ neurons were enriched for neuropeptide signaling pathways, for example, acetylcholine receptor binding (GO:MF; FDR Q = 0.0044) and neuropeptide receptor activity (GO:MF; FDR Q = 1.2 × 10$^{-5}$). This suggests unique epigenetic mechanisms for the regulation of transcription related to receptor signaling in GPe PV+ neurons, but further work is needed to discern these relationships.

## PV+ SNAIL probes generalize to subcortical brain regions in the mouse

Given these complexities, we were interested in the extent to which PV+ enhancer probes chosen from data in one tissue could generalize to other brain regions. Here, we assessed whether SC1 and SC2 SNAIL probes, designed in the cortex, were also selective for PV+ neurons in the striatum and GPe. First, we used cSNAIL ATAC-seq data from the striatum and GPe to model the regulatory sequence properties of PV+ neurons vs. PV- cells in these brain regions, using the population-derived linear SVM strategy (*Figure 4—figure supplement 1*). We assessed the correlation between the score

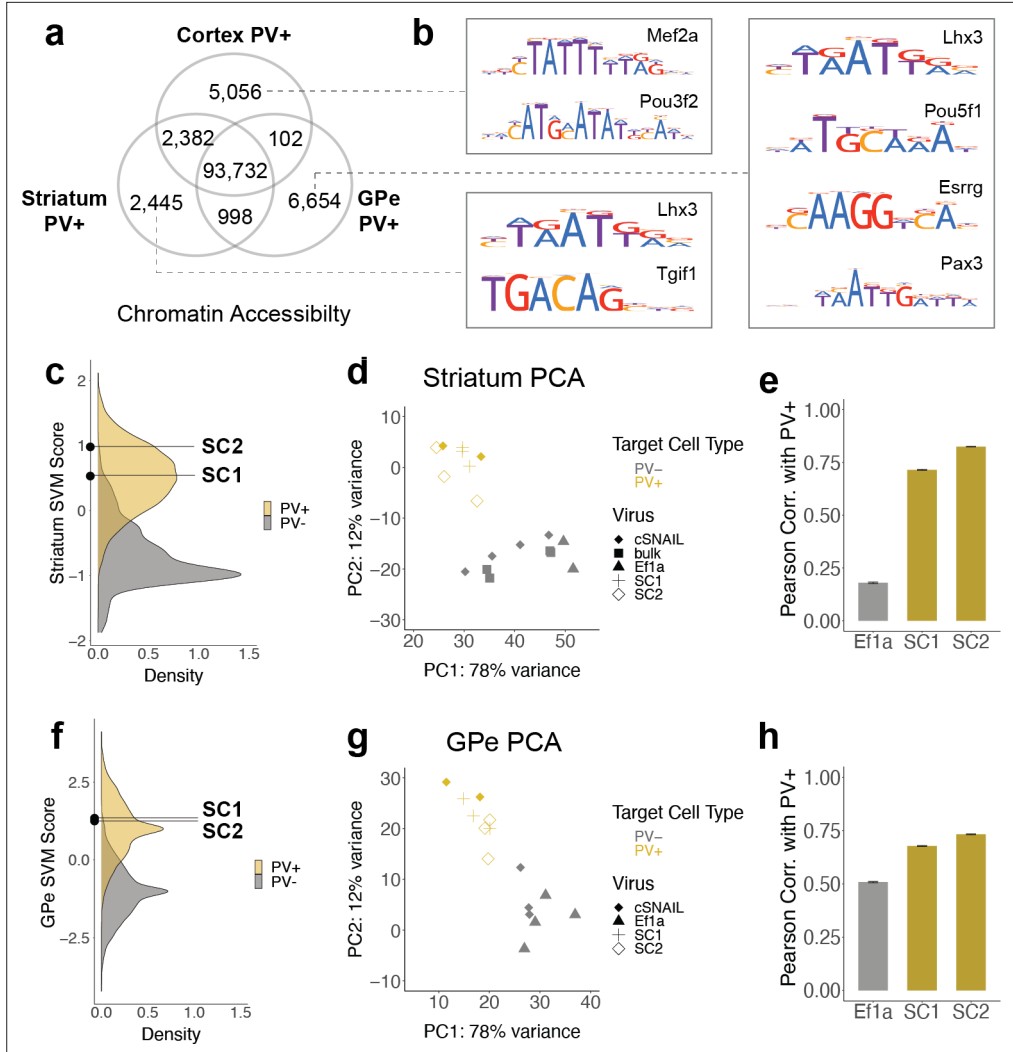

**Figure 4.** SC1 and SC2 generalize to parvalbumin-expressing (PV+) neurons in the striatum and external globus pallidus (GPe). (**a**) Numbers of differential open chromatin regions (OCRs) between PV+ neuron populations in three brain regions (DESeq2 padj<0.01 and |log2FoldDifference| > 1). Brain region-specific OCRs are those that were significantly enriched in that tissue relative to each of the other two tissues. OCRs shared between two brain regions on the Venn diagram are those that were significantly enriched in each of those tissues relative to the excluded tissue. The shared center of the Venn diagram shows all remaining OCRs that have ambiguous or no tissue preference. (**b**) Examples of enriched motifs in brain region-specific PV+ open chromatin relative to all PV+ open chromatin. (**c, f**) Distributions of validation data support vector machine (SVM) scores and SC1 and SC2 scores within striatum and GPe PV+ vs. PV- models. (**d, g**) Principal component analysis (PCA) visualization of ATAC-seq counts in each sample. (**e, h**) Pearson correlation coefficients when comparing the log2 fold difference of cSNAIL PV+ ATAC-seq relative to bulk tissue ATAC-seq and the log2 fold difference of SNAIL ATAC-seq relative to bulk tissue ATAC-seq. Error bars show the 95% confidence intervals.

The online version of this article includes the following source data and figure supplement(s) for figure 4:

**Source data 1.** Differential open chromatin region (OCR) statistics for tissue-specific parvalbumin-expressing (PV+) neuron OCRs in cortex, striatum, and external globus pallidus (GPe).

**Source data 2.** Motif enrichments among tissue-specific parvalbumin-expressing (PV+) neuron open chromatin region (OCR) sequences.

**Source data 3.** Pathway enrichments among tissue-specific parvalbumin-expressing (PV+) neuron open chromatin region (OCR) sequences.

**Source data 4.** Differential open chromatin region (OCR) statistics for parvalbumin-expressing (PV+) Specific Nuclear-Anchored Independent Labeling (SNAIL)-isolated striatal ATAC-seq relative to bulk tissue striatal ATAC-

*Figure 4 continued on next page*

*Figure 4 continued*

seq.

**Source data 5.** Differential open chromatin region (OCR) statistics for parvalbumin-expressing (PV+) Specific Nuclear-Anchored Independent Labeling (SNAIL)-isolated external globus pallidus (GPe) ATAC-seq relative to PV-GPe ATAC-seq.

**Figure supplement 1.** Subcortical parvalbumin (PV)+ vs. PV- support vector machines (SVMs).

**Figure supplement 2.** Comparison of subcortical SC1 and SC2-labeled populations with cortical snATAC-seq cluster markers.

outputs of the cortex PV+ vs. PV- SVM with the striatum PV+ vs. PV- or GPe PV+ vs. PV- SVMs for our set of experimentally identified mouse cortical PV+ neuron enhancers sequences. Enhancer scores were well correlated between the cortex and striatum models (Pearson = 0.73, Spearman = 0.72), indicating shared regulatory sequence determinants between PV+ neurons in the cortex and striatum. There was a low correlation between enhancer scores of the cortex and GPe models (Pearson = 0.25, Spearman = 0.24), indicating differences in the learned PV+ regulatory sequence properties in these regions. SC1 and SC2 sequences were predicted to have PV+-specific activation in both the striatum and GPe (*Figure 4c and f*). However, there were between 1000 and 3000 sequences with more confident PV+ scores than SC1 and SC2 in the striatum and GPe.

We proceeded to isolate SC1- and SC2-labeled cells from these tissues in wild-type mice using Sun1GFP affinity purification and performed ATAC-seq on the tagged populations. We have previously shown high agreement between cSNAIL and Pvalb-2A-Cre labeling in the striatum and GPe (*Lawler et al., 2020*), so we again used cSNAIL ATAC-seq samples from these regions as true PV+ neuron signals. By principal component analysis (PCA), we recovered separation between PV+ samples, including SC1- and SC2-isolated populations, and PV- samples (*Figure 4d and g*). We assessed the correlations between log2FoldDifference in SNAIL and cSNAIL samples, each relative to bulk tissue (striatum) or, where there were no bulk samples available, cSNAIL PV- cells (GPe) (*Figure 4e and h*, *Figure 4—source data 4*, *Figure 4—source data 5*). Pearson correlation coefficients were similar or slightly lower for SC1 and SC2 in the striatum and GPe than for equivalent comparisons in the cortex, indicating less conservation between cSNAIL and SNAIL probe targets (SC1 cortex = 0.85, striatum = 0.71, GPe = 0.68; SC2 cortex = 0.81, striatum = 0.82, GPe = 0.73). Still, these were substantially increased over *Ef1a* correlation with cSNAIL in these tissues, especially for the striatum (*Ef1a* cortex = 0.38, striatum = 0.18, GPe = 0.51).

Finally, by comparing the peak overlaps of SC1 and SC2-enriched OCRs in striatum and GPe with cortical snATAC-seq cluster-specific OCRs, we still identified the PV+ cluster as most similar to SC1 and SC2 cells. As expected, all overlaps in striatum-cortex and GPe-cortex comparisons were lower than those from cortex-cortex comparisons, but the magnitudes of SC1 and SC2 overlapped with the Pvalb cluster in these brain regions were similar to the magnitudes of cSNAIL PV+ overlap with the Pvalb cluster in these brain regions (*Figure 4—figure supplement 2*). In the striatum, the overlaps with the Pvalb cluster were 8% for SC1, 14% for SC2, and 14% for cSNAIL. In the GPe, the overlaps with the Pvalb cluster were 7% for SC1, 7% for SC2, and 9% for cSNAIL. From these interpretations, SC1 and SC2 SNAIL viruses seem to generalize to the striatum and GPe, though they may not be as robust as they are within the cortical context. Additional experimental evidence may be necessary to confirm the appropriateness of PV+ SNAIL viruses for certain applications outside the cortex.

## Esrrg and Mef2 motifs are important for the PV+ neuron-specific activity score of SC1 and SC2 sequences

To interpret the specific sequence patterns within SC1 and SC2 that contribute to their PV+ neuron-specific activity prediction, we assessed commonly used motifs for each model and identified potential matches within the candidate sequences. For all SVMs, we calculated per-base importance scores and hypothetical importance scores for the set of PV+ neuron-specific OCRs that were true positives according to all SVMs (score >0; N = 1755) (*Shrikumar et al., 2019*). Then, for each model, we used TF-MoDISco (*Shrikumar et al., 2018*) to cluster commonly important subsequences called 'seqlets' within these PV+-specific examples. The resulting clusters represent motifs that were high contributors to a positive score in each model. Among the 11 SVMs comparing PV+ neuron open-chromatin

against PV- cells, EXC neurons, VIP+ neurons, or SST+ neurons, we recovered 124 well-supported motifs. Many motifs appeared to be shared across multiple models. Thus, we performed UPGMA clustering on the 124 motifs by sequence similarity using STAMP (*Mahony and Benos, 2007*) and identified 14 motif clusters (*Figure 5a*).

The largest cluster, with 23 motif members, contained representation from all 11 models and had matches to known motifs including the motifs for Esrrg and Rora (*Figure 5—source data 1*). Consistent with an important role for Esrrg in PV+ neurons, *Esrrg* (a.k.a. *Err3*) transcript levels were differentially overexpressed in the PV+ neuron cluster relative the rest of the frontal cortex in snRNA-seq (DropViz subcluster #2–7 Neuron.Gad1Gad2.Pvalb *Esrrg* fold ratio = 8.0, p=1.14 × 10$^{-198}$) (*Saunders et al., 2018*). Esrrg and Rora are key TFs in the Pgc1a transcriptional program, which regulates *Pvalb* expression, mitochondrial function, and transmitter release (*Lin et al., 2005*; *Lucas et al., 2010*). Pgc1a signaling is restricted to PV+ neurons in the brain and may mediate the unique energy demands of fast-spiking neurons (*Paul et al., 2017*; *Lucas et al., 2014*).

The second-largest motif cluster contained 16 motifs, also representing all 11 models, and the motifs had the best matches to motifs for Mef2a, Mef2c, and Mef2d. A cluster of Lhx6-like motifs, a TF necessary for MGE interneuron differentiation from interneuron progenitors (*Liodis et al., 2007*; *Vogt et al., 2014*), was detected with high support from PV+ vs. PV- models and PV+ vs. EXC models, detected with low support from PV+ vs. VIP+ models, and not detected between MGE neuron subtypes PV+ vs. SST+. Interestingly, two clusters of motifs were dominated by PV+ vs. VIP+ activity, including matches for Stat6, Nkx28, and Cux2 motifs. *Cux2* expression is induced by Lhx6 in the MGE, supporting a role in specification of the MGE interneuron lineage (including PV+ and SST+ neurons) from other interneuron lineages (*Zhao et al., 2008*). Overall, these findings indicate both shared and unique sequence properties dictating PV+ neuron-specific regulatory sequence activity relative to other cell types.

SC1 and SC2 represent two experimentally validated PV+ neuron-selective regulatory sequences. To interpret the sequence determinants of their success, we mapped potential motif sites for the 124 TF-MoDISco motifs (*Figure 5—source data 2*) and overlaid these with per-base importance scores for each of the SVMs (*Figure 5—source data 3*). This strategy revealed multiple high-importance subsequences with potential TF binding function. SC1 contained two Esrrg motifs near the sequence center that were high contributors to the PV+ neuron-specific model predictions and matched TF-MoDISco motifs for every model (*Figure 5b*). An additional subsequence with contributions specific to PV+ vs. VIP+ models matched motifs for Sp7. SC2 contained a highly important Mef2 sequence near the center (*Figure 5c*). Additionally, SC2 contained an Esrrg motif with shared importance across all models. Interestingly, the most important features of the SC2 sequence closely resemble those of successful PV+ probe E29 from *Vormstein-Schneider et al., 2020* (*Figure 1—figure supplement 6*). Disruption of Esrrg and Mef2 motifs within SC1 or SC2 resulted in a sharply decreased prediction of PV+ specificity according to the scores across the SVMs (*Figure 5d*). While these impacts are untested in vivo, these analyses provide an intuition for potential nucleotide contributions to PV+ neuron-specific enhancer sequence function in SC1 and SC2.

## Cross-species analyses with SNAIL

The utility of modern cell type-specific technologies often depends on transferability between species, particularly between rodents and primates. Machine learning with SNAIL has the potential to expedite cell-type isolation from multiple species by pre-selecting enhancers with sequence composition that is likely to drive cell type-specific expression across species. To investigate this, we built human PV+ SVMs analogous to the mouse SVMs using single-nucleus open chromatin data from the human cortex (*Corces et al., 2020*). These models learned to discriminate between the sequences of open chromatin peaks in human PV+ neurons relative to human open chromatin peaks in excitatory neurons, VIP+ neurons, SST+ neurons, or the combination of all PV- cells. Although the human data contained fewer cell profiles, models were able to reasonably perform the classification task, with auROC 0.73–0.83 and auPRC 0.68–0.81 (*Figure 6a*). Such models may be useful for predicting enhancer sequence activity in primates, regardless of the species origin of the enhancer sequence. We used these human-trained models to predict PV+ neuron-specific activity for 4428 mouse sequences from experimentally identified PV+ neuron-specific open chromatin peaks. Score outputs from human-trained models were correlated with score outputs from mouse-trained models (*Figure 6b*), suggesting shared PV+

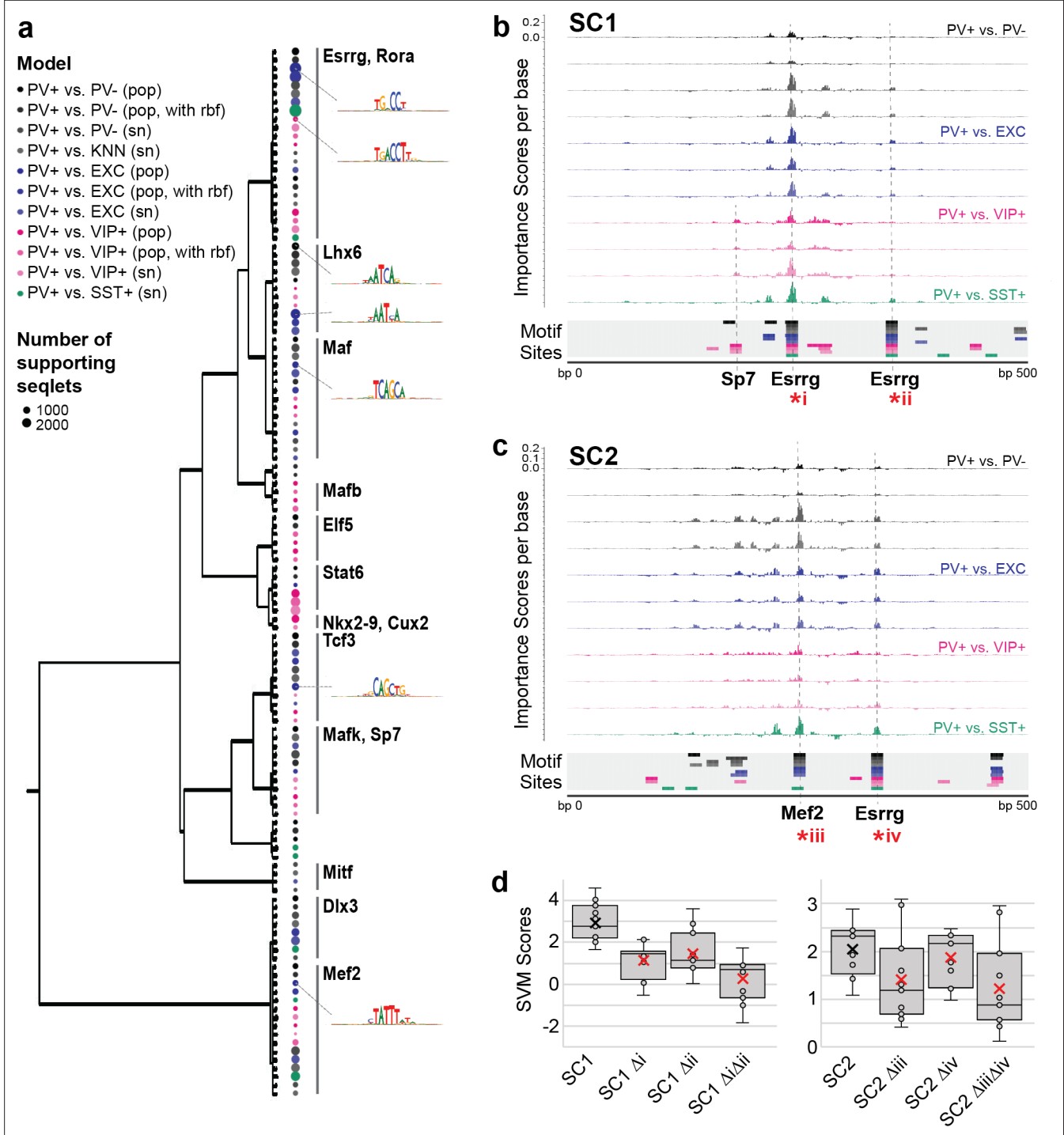

**Figure 5.** Motif interpretation of parvalbumin-expressing (PV+) neuron-specific open chromatin region (OCR) activity. (**a**) Motifs with high contributions to PV+ scores in each support vector machine (SVM), clustered by sequence similarity. The bubble color at each node shows the model that motif was discovered in and the size of the bubble shows the number of seqlets supporting that motif. Clusters are labeled by the clade majority best match for known transcription factor binding motifs. The full list of matches can be found in *Figure 5—source data 1*. (**b, c**) Normalized importance of each base in SC1 (**b**) and SC2 (**c**) sequences for their PV+-specific scores in each SVM. Locations with sequence matches for identified motifs in each SVM (from panel **a**) are shown at the bottom. (**d**) Predicted impacts of motif scrambling on PV+ specificity. Motif mutation sites are shown with asterisks in panels (**a**) and (**b**). Each point is the sequence score from one SVM and 'x' is the mean.

The online version of this article includes the following source data for figure 5:

**Source data 1.** Association of TF-MoDISco motifs to known transcription factor binding motifs.

*Figure 5 continued on next page*

*Figure 5 continued*

**Source data 2.** Identification of motif sites within SC1 and SC2.

**Source data 3.** Normalized per-base importance scores for SC1 and SC2 sequences.

neuron-specific enhancer sequence features between mouse and human PV+ neurons. SC1 was one of the highest-scoring enhancer candidates in both mouse and human SVMs.

Because SC1 was predicted to retain PV+ neuron specificity in primates, we tested the SC1 SNAIL probe in the rhesus macaque cortex via localized intracranial AAV injection. We observed high expression of Sun1GFP with enrichment for PV+ neurons. There was a trade-off between precision and recall of PV+ neuron labeling related to the spatial viral load, with a mean precision of 22% near the injection site center and 67% at the periphery. At the injection center, at least 98% of PV+ neurons were transduced by the SC1 SNAIL probe, while at the periphery, at least 75% of PV+ neurons were transduced. Our preliminary findings are optimistic for PV+ SNAIL probe adoption in primates, but optimal AAV titer should be carefully considered for the specific application.

## Discussion

OCR sequence features provide valuable, underutilized information for cell type-specific enhancer design. Here, we showed that sequence was sufficient to discern the directionality of highly differential OCR activity in different neuron subtypes in most cases. Interpretation of these models revealed rich diversity among the biochemical underpinnings of these classification tasks, reflective of *cis-trans* interactions. The defining sequence properties of cell type-specific OCR activation were robust throughout different data modalities, including ATAC-seq from sorted populations and snATAC-seq, and different classifier types.

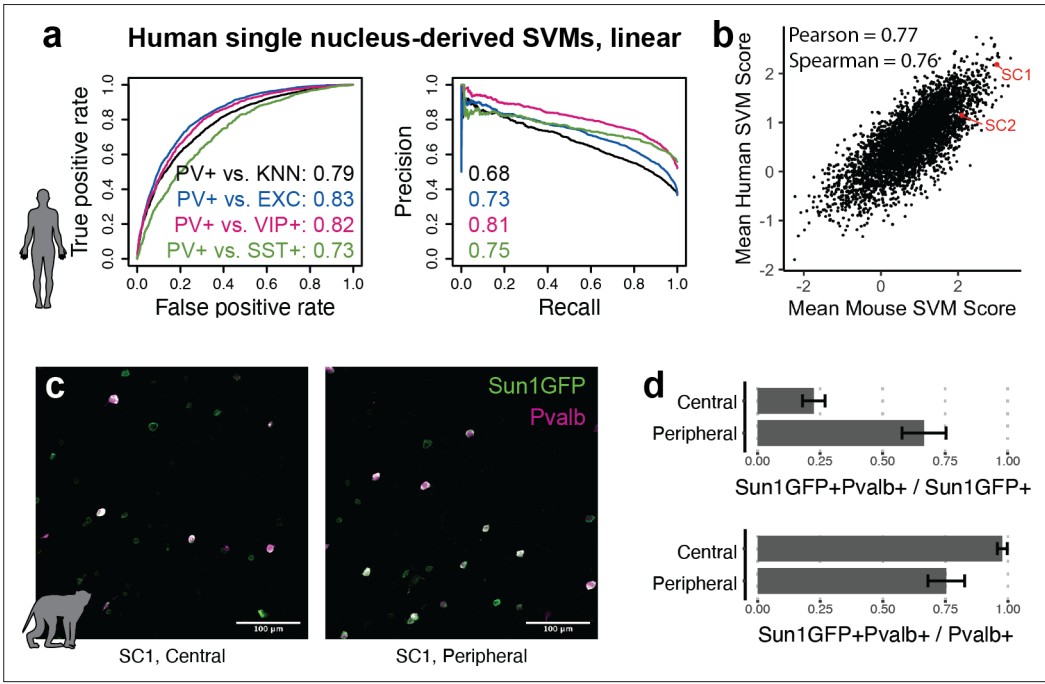

**Figure 6.** Extensions of Specific Nuclear-Anchored Independent Labeling (SNAIL) technologies in primates. (**a**) Receiver operator characteristic and precision-recall performance metrics for parvalbumin-expressing (PV+) support vector machines (SVMs) derived from single-nucleus chromatin accessibility assays of human cortical tissue. The reported numbers are the areas under the curve for each model. (**b**) Comparison of mouse PV+ open chromatin sequences scored by mouse and human SVMs. Axes are the mean SVM scores among the 11 mouse SVMs or 4 human SVMs. (**c**) Images of SC1 AAV activation in the rhesus macaque cortex. (**d**) Quantification of SC1 and Pvalb antibody staining in the macaque cortex. Image sets near the center or peripheral of the injection site were quantified separately.

Thus far, success in cell type-specific AAV creation has been limited to intensive screens of dozens of individual AAV enhancer candidates in which most do not produce highly selective expression (*Vormstein-Schneider et al., 2020*; *Mich et al., 2021*; *Graybuck et al., 2021*). In SNAIL, our framework for cell type-specific AAV engineering, we incorporate machine learning classifiers as an additional filter for improved enhancer pre-selection in order to mitigate the experimental burden. On a set of externally tested PV+ enhancer-driven AAVs (*Vormstein-Schneider et al., 2020*), the average PV+ specificity score across our classifiers was more predictive of PV+ specific AAV expression than the log2 fold difference of snATAC-seq signal, sequence conservation, or accessibility conservation at these loci. With the SNAIL framework, we identified and validated two novel enhancers that drive targeted expression in PV+ neurons in the mouse cortex. While these do not represent enough trials to establish a new conversion rate from cell type-specific OCRs to cell type-specific AAVs, we were encouraged by the immediate success of the first probes we selected. We believe that the incorporation of differential sequence property analyses will continue to improve the throughput of targeted AAV development in new contexts and make cell type-specific enhancer tool development accessible for more researchers.

An additional advantage of incorporating classifiers for cell type-specific enhancer selection is increased interpretability of the factors that govern success. The sequence patterns learned by PV+ models reflected known PV+ neuron biology. Common motifs contributing to successful PV+ probe enhancers included Esrrg, Mef2, and Lhx6, important for the specification and maintenance of the cortical PV+ interneuron lineage (*Zhao et al., 2008*; *Mayer et al., 2018*; *Liodis et al., 2007*). It is interesting to note the varying expression patterns of these TFs. In the cortex, Esrrg expression is mainly restricted to PV+ neurons and Lhx6 is restricted to MGE interneurons, but Mef2 transcripts are widely expressed in many inhibitory and excitatory neurons. It is likely that combinatorial interactions between abundant and cell type-specific TF binding events specify PV+ neuron enhancer activation. The SNAIL framework provides an opportunity to meaningfully leverage these complex sequence codes.

We found that a combination of multiple direct comparisons between the target cell type and other cell types made for particularly useful screening. Here, we used a tiered approach to ensure specific activity at multiple levels of cellular relationships to PV+ neurons. At the broadest level, we modeled PV+ neuron OCR sequences against PV- OCRs, a mixed signature from all other neuron and non-neuron cell types in the mouse cortex. Within neurons, we modeled PV+ vs. EXC neurons, and then PV+ relative to more specific subtypes of inhibitory neurons VIP+ and SST+. Successful SC1 and SC2 sequences contained attributes that made them highly PV+ specific across these comparisons.

SC1-Sun1GFP and SC2-Sun1GFP are new AAV technologies for PV+ neuron labeling and isolation in diverse systems. A unique feature of these viruses is the modified Sun1GFP tag that enables nuclei purification by magnetic beads coated with anti-GFP antibodies. This process is advantageous for isolating genomic and epigenomic signals from the population of interest with no dependence on transgenic strains. In comparison to single-nucleus sequencing technologies, affinity purification with SNAIL is more efficient for addressing targeted hypotheses about a specific cell type. SNAIL may also be paired with single-nucleus sequencing technologies for unprecedented resolution of the substructures within minority cell populations. We took advantage of SNAIL affinity purification to isolate SC1-Sun1GFP and SC2-Sun1GFP nuclei for molecular assessment with ATAC-seq. This represents a novel approach for validating new cell type-specific AAVs. We found that SC1 and SC2 PV+ SNAIL probes had high molecular agreement with cells tagged in the Pvalb-2A-Cre mouse strain, making them a reasonable alternative to transgenic strain technology. In addition to their success in the intended brain region (cortex), SC1 and SC2 PV+ SNAIL viruses also generalized to subcortical regions, the striatum, and GPe enough to isolate PV+ ATAC-seq signal. Moreover, the SC1 enhancer has sequence characteristics consistent with primate PV+-specific activation and has biased expression in PV+ neurons in the macaque brain. It will be interesting to conduct further comparisons of the composition of cell populations that are labeled by these tools in different brain regions and species. We speculate that the SNAIL framework could be strategically employed to design tools for specific subtypes of PV+ neurons or other cell populations that may be brain region or species specific.

In general, pairing cell type-specific enhancers with AAVs provides much more flexibility and scalability than transgenic technologies. However, there are drawbacks to certain applications. AAVs require time to reach peak expression, usually 2–4 weeks, although some may be robust earlier. This

means they are not appropriate for developmental studies in very young animals. These and other developmental time points outside of the scope of the training data may have unintended effects on enhancer expression and specificity. Additionally, there are limitations to the transduction efficiency, so AAVs may not be ideal for studies where it is important to label all cells of a certain type. Finally, an enhancer's activity in AAV may differ from the same enhancer's activity in the host genome due to the surrounding genomic and epigenomic context. Enhancer activity may also fluctuate in response to different conditions because enhancers are dynamic actors in the regulation of gene expression. However, machine learning model-based prioritization of characteristic sequences may minimize this risk.

Excitingly, there are many opportunities for extensions of the SNAIL framework that enable cell type-specific interrogation in unprecedented settings. In a direct application that we began to explore in this work, SNAIL probes could be used to isolate cell types of interest in diverse species. High evolutionary conservation in the rules for enhancer activity has been demonstrated by the success of machine learning models in predicting enhancers across mammals (*Minnoye et al., 2020*; *Kelley, 2020*; *Kaplow et al., 2020*; *Chen et al., 2018*). Models trained across multiple species could further improve the transferability of probes across species. For example, a new approach that explicitly encourages the model not to learn signatures of species-specific enhancer activity might be especially promising for designing cross-species SNAIL probes (*Cochran et al., 2021*).

Beyond the potential ability for cross-species probe design, the SNAIL framework provides the opportunity to develop viral tools targeting previously unexplored cell types that are identifiable in snATAC-seq both within and across species. There is potential to divide subpopulations at multiple levels and design extremely specific technologies, including specialization for neuron populations within a specific brain region. Other applications may exploit changes in enhancer sequence activity in disease and other contexts to target specific cell states. In addition to creating cell type-specific enhancers to isolate cell types of interest, machine learning model-selected enhancer sequences may be used to drive the expression of a gene for cell type-specific circuit manipulation, as has been achieved with channelrhodopsin and DREADDS (*Lee et al., 2010*; *Vormstein-Schneider et al., 2020*). This approach could also be used to overexpress a particular ion channel, neurotransmitter receptor, gene variant, or guide RNA for a CRISPR-based gene manipulation strategy. We anticipate that SNAIL will provide a foundation for using sequence signatures underlying cell type-specific enhancer activity to develop tools for understanding cell type-specific function of a variety of cell types in diverse contexts.

# Materials and methods

**Key resources table**

| Reagent type (species) or resource | Designation | Source or reference | Identifiers | Additional information |
|---|---|---|---|---|
| Strain, strain background (*Mus musculus*) | Pvalb-2A-Cre | Jackson Laboratories | Stock # 012358 | B6.Cg-Pvalb[tm1.1(cre)Aibs] |
| Strain, strain background (*M. musculus*) | Ai14 | Jackson Laboratories | Stock # 007914 | B6.Cg-Gt(ROSA)26Sor[tm14(CAG-tdTomato)Hze]/J |
| Recombinant DNA reagent | pAAV-Ef1a-DIO-Sun1GFP-WPRE-pA | Addgene | Plasmid #160141 | cSNAIL vector from *Lawler et al., 2020* |
| Recombinant DNA reagent | pAAV-Ef1a-Sun1GFP-WPRE-pA | This paper | | *Ef1a* vector (+ control) |
| Recombinant DNA reagent | pAAV-SC1-Sun1GFP-WPRE-pA | This paper | | SC1 vector targeting PV+ neuron expression |
| Recombinant DNA reagent | pAAV-SC2-Sun1GFP-WPRE-pA | This paper | | SC2 vector targeting PV+ neuron expression |
| Recombinant DNA reagent | pAAV-neg-Sun1GFP-WPRE-pA | This paper | | Negative control vector |
| Antibody | Anti-Pvalb (rabbit polyclonal) | Swant | Cat# PV27 | IF (1:750) |

*Continued on next page*

*Continued*

| Reagent type (species) or resource | Designation | Source or reference | Identifiers | Additional information |
|---|---|---|---|---|
| Antibody | Anti-Pvalb (mouse monoclonal) | Swant | Cat# 235 | IF (1:1000) |
| Antibody | Anti-GFP (rabbit polyclonal) | Invitrogen | Cat# A-11122 | IF (1:1000) |
| Antibody | Anti-GFP (rabbit monoclonal) | Invitrogen | Cat# G10362 | Affinity purification (1:150) |
| Software, algorithm | ls-gkm | *Lee, 2016* | https://github.com/Dongwon-Lee/lsgkm | SVM training |
| Software, algorithm | Keras | https://keras.io | | CNN training |

## Experimental design

We designed the ATAC-seq experiments to meet the current ENCODE standards of two or more biological replicates (i.e., individuals) per condition, each consisting of 50 million non-duplicate non-mitochondrial paired-end sequencing reads. The initial cSNAIL experiments to define candidate PV+ enhancers were performed on primary motor cortex and isocortex samples in triplicate on female mice aged 2–3 months old. All subsequent cSNAIL and SNAIL molecular experiments for the validation of PV+ SNAIL probes were performed in the cortex, striatum, and GPe with two or three biological replicates. Each of these cohorts included at least one male and one female mouse, all 2–4 months old. Control samples for SNAIL comparisons included cSNAIL PV+, cSNAIL PV-, and cells labeled by the *Ef1a*-Sun1GFP virus. Details for all experiment samples can be found in *Supplementary file 1*. Data primary to this publication can be accessed through the NCBI Gene Expression Omnibus (https://www.ncbi.nlm.nih.gov/geo/), accession number GSE171549.

## Nuclei isolation for ATAC-seq

ATAC-seq data were generated using an affinity purification approach with cSNAIL or SNAIL to isolate PV+ neurons from the mouse isocortex, as described in *Lawler et al., 2020*. Briefly, mice were overdosed with isoflurane, decapitated, and rapidly dissected. Fresh brain tissue was sectioned coronally on a vibratome for precision, and we dissected brain regions relevant to the specific experiment to be processed as separate samples. All dissections took place in cold, oxygenated artificial cerebrospinal fluid (aCSF). After dissection, we isolated nuclei from the samples by 30 strokes of Dounce homogenization with the loose pestle (0.005 in clearance) in lysis buffer as described in *Buenrostro et al., 2015a*. The nuclei were filtered through a 70 µm strainer and pelleted with 10 min of centrifugation at $2000 \times g$ at 4°C. We resuspended the nuclei pellets in wash buffer (0.25 M sucrose, 25 mM KCl, 5 mM $MgCl_2$, 20 mM Tricine with KOH to pH 7.8, and 0.4% IGEPAL) for the affinity purification steps.

## Affinity purification of Sun1GFP+ and Sun1GFP- nuclei

The nuclei suspension was incubated with anti-GFP antibody (Invitrogen, Carlsbad, CA; #G10362) in wash buffer for 30 min at 4°C with end-to-end rotation. After this period, we added Protein G Dynabeads (Thermo Fisher Scientific, Waltham, MA; Cat# 10004D) to the reaction and incubated again for 20 min. We separated the Sun1GFP+ fraction from the Sun1GFP- fraction on a magnetic bead rack. Sun1GFP- nuclei in the supernatant were centrifuged at $2000 \times g$ for 10 min to pellet nuclei, washed one time, and filtered with a 40 µm cell strainer. The Sun1GFP+ nuclei attached to the beads were washed 3–4 times with 800 µL wash buffer by resuspending the sample, letting it settle onto the magnet, and removing the buffer. Where cell yield was not a concern, we also performed a large-volume wash with 10 mL wash buffer and filtered through a 20 µm cell strainer. All nuclei preparations were resuspended in water for the ATAC-seq reaction.

## ATAC-seq library construction

For each sample, a small aliquot was stained with DAPI (Thermo Fisher Scientific; Cat# 62248) and the concentration of nuclei was determined by counting DAPI+ nuclei with a hemocytometer. Next, we combined 50,000 nuclei, 25 µL Tagment DNA Buffer, and 2.5 µL Tagment DNA Enzyme I (Illumina, San Diego, CA; Cat# 20034198) into 50 µL total for the transposition reaction. The reaction was incubated

at 37°C for 30 min with 300 rpm mixing. Samples containing beads were gently resuspended every 5–10 min throughout the incubation to prevent the beads from staying settled at the bottom. Immediately following incubation, the DNA was column purified with the QIAGEN MinElute PCR Purification kit (QIAGEN, Hilden Germany; Cat# 28004). Libraries were amplified to ⅓ saturation with dual-indexed Illumina primers (*Preissl et al., 2018*). We ensured that samples had the characteristic periodic fragment length distribution of high-quality ATAC-seq using TapeStation assessment (Agilent Technologies, Santa Clara, CA). Successful samples were sequenced at low depth on the Illumina MiSeq system to determine appropriate library pooling and sequencing depth, then paired-end sequenced for 2 × 150 cycles with the Illumina NovaSeq 6000.

## Animal use

All animals for ATAC-seq experiments were either wild-type mice (C57BL/6J; Jackson Laboratory, Bar Harbor, ME; Stock # 000664) for SNAIL experiments or heterozygous Pvalb-2A-Cre mice (B6.Cg-Pvalb$^{tm1.1(cre)Aibs}$/J; Jackson Laboratory; Stock # 012358) (*Madisen et al., 2010*) on a C57BL/6J background for cSNAIL experiments. Imaging animals were either Pvalb-2A-Cre or double transgenic Pvalb-2A-Cre/Ai14 (Ai14 strain; B6.Cg-Gt(ROSA)26Sor$^{tm14(CAG-tdTomato)Hze}$/J; Jackson Laboratory; Stock # 007914). All mice were 2–4 months old at the time of the tissue experiments. Initial PV+ cSNAIL data for creating the sorted cell PV+ vs. PV- model was collected from female mice, but all subsequent validation experiments included representation from both sexes. All animals were housed with a 12 hr light cycle, and experiments were performed 2–3 hr after lights on. Animals for the data primary to this study received no treatments other than the retro-orbital AAV injections. However, previously published cSNAIL data used in the analysis included healthy animals that received stereotaxic saline injections to the medial forebrain bundle (*Lawler et al., 2020*).

## Molecular cloning

To make the nonspecific control viral vector pAAV-Ef1a-Sun1GFP, we made modifications to pAAV-Ef1a-Cre with restriction enzyme cloning. pAAV-Ef1a-Cre was a gift from Karl Deisseroth (Addgene, Watertown, MA; plasmid #55636; http://n2t.net/addgene:55636; RRID:Addgene_55636). First, we added a multiple cloning site before the *Ef1a* promoter to create easy promoter swapping for later use. The multiple cloning site insert was synthesized by Integrated DNA Technologies, Coralville, IA, and was inserted between BshTI and MluI sites upstream of the *Ef1a* promoter. Next, we used BamHI and EcoRI sites to replace the Cre gene with a modified Sun1GFP gene identical to the one in our cSNAIL technologies.

The resulting pAAV-Ef1a-Sun1GFP vector was further modified to create the other constructs. The PV+ SNAIL probes were designed to contain one PV+-specific enhancer candidate sequence, a synthetic intron for RNA stabilization, the Sun1GFP gene, a WPRE signal, and a polyA signal. From pAAV-Ef1a-Sun1GFP, the *Ef1a* promoter and intron region was removed and replaced with the sequence for a PV+-specific enhancer candidate and the synthetic intron. Inserts for SC1 and SC2 were synthesized by Integrated DNA Technologies and cloned into the vector using restriction sites for NdeI and BamHI. To ensure that no expression was being driven from the synthetic intron sequence itself, we similarly cloned a negative control construct containing the synthetic intron, but no enhancer candidate sequence. All transformations during cloning were performed in MegaX DH10B cells (Invitrogen, #C640003) and confirmed with Sanger sequencing.

## AAV PHP.eB production for mice

AAV was produced in AAVpro(R) 293T cells (Takara, Kyoto, Japan; #632273) by co-transfection of the genome pAAV, an AAV helper plasmid, and pUCmini-iCAP-PHP.eB. pUCmini-iCAP-PHP.eB was a gift from Viviana Gradinaru (http://n2t.net/addgene:103005; RRID: Addgene 103005) (*Chan et al., 2017*). The AAV particles were precipitated with polyethylene glycol (PEG 8000, Sigma-Aldrich, St. Louis, MO; Cat# P2139-500G) and purified on an iodixanol gradient (OptiPrep, Sigma-Aldrich, Cat# D1556-250ML) with ultracentrifugation for 2.5 hr at 350,000 × *g* at 18°C. We filtered and concentrated the virus in PBS using Amicon Ultra-15 centrifugation filters (Millipore, Burlington, MA; #UFC905024). The viral titer was measured with the AAVpro(R) Titration Kit (Takara, #6233) and AAV was stored single-use aliquots at –80°C until injection.

## AAV9-2YF production for primates

AAV vectors were produced in 293AAV cells (Cell Biolabs; Cat# AAV-100) using a triple transfection method (*Grieger et al., 2006*). Recombinant AAVs were purified by iodixanol gradient ultracentrifugation, buffer exchanged, and concentrated with Amicon Ultra-15 Centrifugal Filter Units (Cat# UFC8100) in DPBS and titered by using QuickTiter AAV Quantitation Kit (Cell Biolabs; Cat# VPK-145) and quantitative PCR relative to a standard curve.

## Mouse AAV delivery

Animals were anesthetized with 2–3% isoflurane until no pedal withdrawal reflex was observed. Then, we injected $4 \times 10^{11}$ vg total (50 µL) of virus into the retro-orbital cavity and treated the eye with 0.5% proparacaine hydrochloride ophthalmic solution. The animals were monitored while the virus incubated for 3–4 weeks until endpoint experiments.

## Primate AAV delivery

All animal procedures were in accordance with the National Institutes of Health Guide for the Care and Use of Laboratory Animals and approved by the University of Pittsburgh's Institutional Animal Care and Use Committee (IACUC) (Protocol ID 19024431). Monkey B was a 3-year-old male (4.6 kg). The monkey was sedated with ketamine (15 mg/kg IM) and transported to a surgery room. The animal was kept anesthetized with isoflurane. The brain was placed and fixed in a stereotaxic frame (Kopf Instruments). We shaved, cleaned the brain, and removed the calvarium above the region of interest. We cut open the dura and exposed the brain. We injected 20 µL PV58 (R58) around Brodmann areas 4 and 6 using Harvard Apparatus infusion pump.

## Mouse brain immunofluorescence

Tissues were fixed with whole-body 4% paraformaldehyde (PFA) perfusion, and the brains were incubated in 4% PFA for an additional 12–24 hr after dissection. Coronal slices 80 µm thick were made with a vibratome. Free-floating sections were stained for parvalbumin with Pvalb (Swant, Marley, Switzerland; PV27) primary antibody with Alexa Fluor 405 (Invitrogen, #A-31556) or Alexa Fluor 594 (Cell Signaling Technology, Danvers, MA; #8889) secondary antibodies.

## Primate brain immunofluorescence

The animal was sedated with ketamine (15 mg/kg IM) and kept anesthetized with isoflurane. We removed the calvarium first and then perfused the monkey through the circulatory systems with 4 L of ice-cold aCSF (124 mM NaCl, 5 mM KCl, 2 mM $MgSO_4$, 2 mM $CaCl_2$, 23 mM $NaHCO_3$, 3 mM $NaH_2PO_4$, 10 mM glucose; pH 7.4, osmolarity 290–300 mOsm) oxygenated with 95% $O_2$:5% $CO_2$. We then cut open the dura and extracted the brain. We partitioned the cortical tissue into blocks, which were drop-fixed in 4% PFA for 48 hr and then changed through a series of three graded sucrose solutions. Sections (40 µm) were cut on a cryostat and stored at 4°C in a solution of 30% glycerol/30% ethylene glycol. Sections were rinsed three times in 0.1 M PB and treated with 0.1% $NaBH_4$ for 30 min at room temperature (RT). The tissue was rinsed eight times in PBS-B, treated with 0.3% Triton (30 min at RT), and placed in 20% normal donkey serum blocking solution (Jackson, 017-000-121) for 2 hr at RT. Sections were subsequently incubated for 72 hr at 4°C using parvalbumin (Swant, PV, mouse, 235, 1:1000) and green fluorescent protein (Invitrogen, GFP, rabbit, A-11122, 1:1000) primary antibodies. Tissue was rinsed four times in PBS-B and incubated for 24 hr at 4°C in secondary antibodies (donkey) conjugated to Alexa 488 (Invitrogen, anti-rabbit, 1:500, A21206) or Alexa 568 (Invitrogen, anti-mouse, 1:500, A10037). Sections were rinsed three times in PBS-B. Tissue was mounted using ProLong Diamond Antifade mountant (Thermo Fisher, P36970), sealed with #1.5 coverslips, and stored at 4°C until imaging.

## Microscopy

Images were taken of the motor cortex with laser scanning confocal microscopy. Cells were counted in each channel with Fiji (*Schindelin et al., 2012*) and assigned as double-labeled or single-labeled manually. Individual images from 1 to 3 animals were treated as replicates to determine the mean and standard error of the mean for specificity and efficiency quantifications.

## ATAC-seq data processing

Samples were processed from the paired-end FASTQ files using the ENCODE ATAC-seq pipeline (https://github.com/ENCODE-DCC/atac-seq-pipeline; **ENCODE DCC, 2022**) with the following changes from default behaviors: atac.cap_num_peak = 300000, atac.idr_thresh = 0.1. All samples had high TSS enrichment (>15) and clear periodicity, indicative of good data quality. Optimal IDR peaks were determined for biological replicates of the same cell type, brain region, and sequencing batch (https://github.com/kundajelab/idr; **Li et al., 2011**). IDR peaks were then merged to define the combined peak regions (OCRs) for each analysis using bedtools (**Quinlan and Hall, 2010**). Specifically, we defined sets of OCRs for (i) cortex PV+ and PV- cSNAIL samples, (ii) PV+, EXC, and VIP+ INTACT samples (**Mo et al., 2015**), and (iii) cortex, striatum, and GPe bulk samples, PV+ and PV- cSNAIL samples, SC1-Sun1GFP samples, SC2-Sun1GFP samples, and *Ef1a*-Sun1GFP samples. We constructed count tables including the relevant samples on each of these OCR backgrounds using Rsubread featureCounts version 1.28.1 (**Liao et al., 2019**). These three count tables were used to form the basis of (i) the sorted population PV+ vs. PV- models, (ii) the sorted population PV+ vs. EXC and PV+ vs. VIP+ models, and (iii) analysis of SC1 and SC2 SNAIL PV+ probes in the cortex, striatum, and GPe, respectively.

The counts were modeled using the negative binomial distribution in DESeq2 (**Love et al., 2014**). We assessed the coefficient of cell group, where cell groups were unique tissue, virus, cell-type combinations, and we controlled for sex differences where both were present: DESeq2 design ~sex + cellGroup. Differential peaks were defined strictly for applications i and ii related to building models (padj<0.01 and |Log2FoldDifference| > 1) and more loosely for application iii to compare across viruses (padj<0.05 and |Log2FoldDifference| > 0.5). Related to **Figure 3**, only cortical samples from count matrix iii were included in the DESeq2 model, while the **Figure 4** DESeq2 model included samples from all three brain regions.

## snATAC-seq processing

We downloaded the following samples of snATAC-seq from the mouse MOp in Snap file format from http://data.nemoarchive.org/biccn/ (RRID:SCR_016152): CEMBA171206_3C, CEMBA171207_3C, CEMBA171212_4B, CEMBA171213_4B, CEMBA180104_4B, CEMBA180409_2C, CEMBA180410_2C, CEMBA180612_5D, and CEMBA180618_4D (**Li et al., 2021**). We processed these samples using SnapATAC version 1.0.0 (**Fang et al., 2021**). We restricted the analysis to nuclei that passed filtering as defined by the original authors (**Li et al., 2021**). This removed nuclei that had fewer than 1000 reads, TSS enrichment < 10, or doublet signatures detected by Scrublet (**Wolock et al., 2019**). Filtered samples contained 6700–10,983 nuclei each, making a total of 78,525 nuclei. We transformed the data to a count matrix over 5000 bp bins and combined the snap objects. We removed bins overlapping with the ENCODE blacklist, mitochondrial regions, and the top 5% of bins that overlapped with invariant features. We reduced dimensionality and selected 18 significant components. We used these to correct for batch effects with Harmony (**Korsunsky et al., 2019**). We performed Louvain clustering on the outputs from Harmony using runCluster() with the option louvain.lib="R-igraph".

We assigned cell types to clusters by accessibility at promoters and gene bodies of marker genes (**Figure 1—figure supplement 1**) and by comparison to the cell annotations from the original authors (**Li et al., 2021**). We called peaks for each cluster using MACS2 with the options --nomodel --shift 0 --ext 73 --qval 1e-2 -B --SPMR --call-summits (**Zhang et al., 2008**). We merged overlapping peaks across all clusters, resulting in 415,813 total OCRs. We defined differential OCRs using the findDAR() function with test.method = "exactTest" and were required to meet padj<0.01 (Benjamini–Hochberg corrected) and |log2FoldDifference| > 1. For comparisons to groups of clusters, for example, PV+ vs. EXC, we performed separate tests for PV+ vs. each excitatory cluster and selected the intersection of differential OCRs.

For the human models, we downloaded human snATAC-seq data from the parietal, temporal, and frontal cortex (**Corces et al., 2020**) from the NCBI Gene Expression Omnibus, accession number GSE147672. We assigned the cluster identities assigned in the original data publication (**Corces et al., 2020**). We defined differential peaks using the ArchR package (**Granja et al., 2021**) with the cutoffs padj<0.01 (Benjamini–Hochberg corrected) and |log2FoldDifference| > 1.

## SVM data preparation

SVMs were developed to predict the direction of differential activity from sequences underlying differential OCRs between two cell types or groups of cell types. Because ATAC-seq summit regions are highly enriched for TF binding motifs, we centered on the peak summits within differential ATAC-seq OCRs and extended in both directions for a total fixed sequence length of 500 bp, a convenient length for AAV cloning. Peak summits were defined by MACS2 (*Zhang et al., 2008*), and only summit regions of peaks called within the cell type of interest were retained. For data from sorted cells, we used optimal IDR peaks across biological replicates of the given cell type. For example, in a PV+ vs. VIP+ model comparison, the positive model input examples were 500 bp summit-centered regions of PV+ IDR peaks that overlapped PV+-specific differential OCRs and the negative model input examples were 500 bp summit-centered regions of VIP+ neuron IDR peaks that overlapped VIP+ neuron-specific differential OCRs. For snATAC-seq data, we used peaks called within a cluster to define the relevant summit regions. If multiple cell clusters were involved in the comparison, for example, the excitatory neuron vs. inhibitory neuron model, we used summits found in any peak set from a cluster within that category. In cases where there were multiple summits within a differential OCR, all summits greater than 100 bp apart from each other were retained.

After defining the genomic locations of the summit-centered differential OCRs, we did additional filtering to finish preparing the data for model training. First, we restricted the data to enhancer regions because they usually have more cell type specificity than promoters and may be governed by different sequence properties. Therefore, we filtered out regions that were within 2000 bp of a TSS using RefSeq annotations downloaded from the UCSC Table browser in July 2020 (*Kuhn et al., 2013*). Next, for mouse models, we removed super-enhancers because they also may be governed by different sequence features and are not useful for AAV probe design because they are too large. We downloaded mm9 coordinates of mouse cortex super enhancers defined by H3K27ac from the dbSuper database (*Khan and Zhang, 2016*) and converted these to mm10 coordinates using UCSC liftOver with minmatch = 0.95 (*Kuhn et al., 2013*). Using bedtools intersect (*Quinlan and Hall, 2010*), we removed regions with any super-enhancer overlap. Finally, we used bedtools getfasta (*Quinlan and Hall, 2010*) to retrieve the sequences at these genomic coordinates from the mm10 assembly downloaded from UCSC genome browser in May 2018 (*Kuhn et al., 2013*) or the hg38 genome assembly GCA_000001405.27 downloaded from NCBI, and we removed any sequences that contained uncertain bases (Ns).

## SVM model construction

We divided sequences into separate partitions by chromosome for model training, validation, and final testing. For mouse models, the training sets included chromosomes 3–7, 10–19, and X, the validation sets included chromosomes 8 and 9, and the test sets included chromosomes 1 and 2. For human models, the training sets included chromosomes 3–7, 10–22, and X, the validation sets included chromosomes 8 and 9, and the test sets included chromosomes 1 and 2. We input the training data into LS-GKM's gkmtrain, and we evaluated model performance with gkmpredict (*Lee, 2016*). Because the input data was summit-centered, all models used the center-weighted gkm kernel, option -t 4, or the center weighted gkm rbf kernel, option -t 5. We tuned the -l, -k, -d, -c, and -w parameters for word length, number of informative columns, number of mismatches to consider, regularization, and class-weighted regularization, respectively, to maximize the validation set F1 scores. We used default values for other parameters. We calculated auROC and auPRC metrics and visualized on training, validation, and test sets using the ROCR package in R (http://ipa-tys.github.io/ROCR/; *Sing et al., 2020*). All paper figures reflect final test set performance. The details of all parameter settings and performance metrics of the final models are reported in *Figure 1—source data 1*.

## CNN data preparation

We conducted differential accessibility analysis using DESeq2 (*Love et al., 2014*) to identify regulatory regions that display cell type-specific accessibility in ATAC-seq in PV+ neurons relative to other background cell types (PV-, VIP+, EXC). We used PV+ and PV- neuron ATAC-seq samples generated in this study as well as PV+, VIP+, and EXC neuron ATAC-seq samples from *Mo et al., 2015*. To conduct differential accessibility analysis, we obtained genomic coordinates of all 200 bp bins in the mm10 reference genome, starting from the 200 bp bin at the beginning of each chromosome and including

all following contiguous nonoverlapping 200 bp bins. We then filtered out any bin that overlaps with an artifact region (*Amemiya et al., 2019*) or with regions that have unknown nucleotides (obtained from the UCSC twoBitInfo utility using the -nBed option). During this step, regions near the ends of chromosomes were filtered out. Then, using the featureCounts function in the subread package (*Liao et al., 2014*), we counted the reads mapping to each of the 200 bp bins in the ATAC-seq samples obtained from every included ATAC-seq sample. We then use the DESeq2 R package (*Love et al., 2014*) to identify bins that were differentially accessible between (i) PV+ and PV-, (ii) PV+ and VIP+, and (iii) PV+ and EXC neurons at a Benjamini–Hochberg FDR-adjusted p-value cutoff of 0.01. For each of the three comparisons, significant differential bins that displayed PV+ specificity (log2Fold-Difference > 0) were used as positive examples for CNN training and significant differential bins that displayed negative log2FoldDifference (log2FoldDifference < 0) were used as negative examples for CNN training.

## CNN model construction

We trained three separate CNN models that relate sequence to comparative regulatory activity (*Zhou and Troyanskaya, 2015*; *Kelley et al., 2016*; *Quang and Xie, 2016*). For each significant differential 200 bp bin, we obtained the 1000 bp sequence surrounding the center of the bin from the mm10 reference genome and trained the CNN to predict the positive or negative class label. We held out sequence examples underlying all significant differential bins on chromosome 4 as a validation set to evaluate hyperparameter settings and to choose the best performing final model. We also held out sequence examples underlying all significant differential bins on chromosomes 8 and 9 as a test set for final evaluation. Because we had different validation and test sets from those used for the SVM, we did not use any results from the SVM to influence our approach to designing the CNN architecture or any other aspects of CNN training. We implemented our CNN model in Keras 2.2.4 (https://keras.io/) with a theano backend (*Al-Rfou et al., 2016*). We created a one-hot encoded representation of the sequence, a 4 × 1000 binary matrix representing positions and occurrences of the 4 nucleotide characters (A,T,G and C) on the sequence, which was propagated through the network. Our CNN architecture consisted of multiple layers of convolution kernels stacked on top of each other (*Figure 1—figure supplement 3*). The first such layer consisted of 1000 convolution kernels, each with a kernel width of 8 and height of 4, which scan the input sequence in chunks of 8 nucleotides. We applied rectified linear unit (ReLu) activations on the outputs of these convolution kernels. This initial layer is followed by a variable number of convolution layers with the same number of kernels (100), each of width 8 and height 1. We applied ReLu activations on these convolution outputs as well. These convolution layers are then followed by a set of max pooling operations that selects the maximum value from a set of 13 adjacent units (pooling size = 13). We set the stride for the max pooling operation to 13 units, meaning that it selected the maximum values from contiguous chunks of 13 adjacent outputs from the previous layer. We applied dropout regularization (*Srivastava et al., 2014*) on the outputs of the max pooling operation to prevent overfitting to the training set. We then flattened the outputs of the max pooling layer into a single vector and passed them to a single output unit with a sigmoid activation function. We used stochastic gradient descent (SGD) to minimize binary cross-entropy loss (log loss) between the output of this unit and the positive/negative class label to learn model parameters.

Each model was trained for 100 passes through the training set (or 'epochs'). For the PV+ vs. PV- and the PV+ vs. VIP+ tasks, we evaluated model performance and chose the best performing model based on the value of the binary cross-entropy loss on the validation set. For the PV+ vs. EXC task, we chose the final model based on a combination of auROC and auPRC on the validation set. We ignored small differences in validation auROC and auPRC (± 0.02) while selecting the final PV+ vs. EXC model. Tuning only the number of variable convolution layers (0, 1, or 2), and the dropout probability for the max pooling output (0.2, 0.4, or 0.5), we were able to achieve strong auROCs and auPRCs on the held-out validation sets. Therefore, we did not attempt to vary learning rate for SGD (0.01), momentum (0.0), batch size (30), number of training epochs (100), number of filters in the first convolution layer (1000), number of filters in subsequent convolution layers (100), kernel sizes (8), max pooling size (13), and stride (13). A table of hyperparameter settings and associated performance metrics (loss value, auROC, auPRC) on training, validation, and test sets is provided in *Figure 1—source data 2*.

## Broad promoter sequences

The sequences of *Gfap*, *Camk2a*, and *Dlx* promoters (*Figure 1—figure supplement 2*) were extracted from AAV plasmids with confirmed cell type-specific activity in vivo. The *Gfap* promoter sequence was from hGFAP-GFP (Addgene plasmid #40592; http://n2t.net/addgene:40592; RRID:Addgene_40592). The *Camk2a* promoter sequence was from pENN.AAV.CamKII0.4.eGFP.WPRE.rBG (Addgene plasmid #105541; http://n2t.net/addgene:105541; RRID:Addgene_105541). The *Dlx* promoter sequence was from pAAV-mDlx-GFP-Fishell-1 (Addgene plasmid #83900; http://n2t.net/addgene:83900; RRID:Addgene_83900) (*Dimidschstein et al., 2016*).

## SVM score analysis for external PV+ AAV screen

Externally tested PV+ AAV enhancer sequences (*Vormstein-Schneider et al., 2020*) were scored through all cortical PV+ SVMs. To enable comparison between models, scores were normalized to standard deviations from 0 using the standard variation of the validation data set for each model. For each pair of models, the sequence scores were assessed for correlation with cor() function from the R Stats package (https://www.rdocumentation.org/packages/stats/versions/3.6.2) with the Pearson method and visualized using the corrplot package in R (https://github.com/taiyun/corrplot; *Taiyun, 2022*) (*Figure 1—figure supplement 4*).

## Alternative prioritization explorations for external PV+ AAV screen

Common alternative approaches for prioritizing enhancer candidates for cell type-specific AAV design include log2FoldDifference and conservation-based ranking. We show that machine learning models are more predictive of success than these approaches by evaluating on the external PV+ enhancer AAV screen (*Vormstein-Schneider et al., 2020*). The log2FoldDifference of ATAC-seq signal in different cell-type comparisons was evaluated from snATAC-seq data (*Li et al., 2021*). We added the exact genomic locations of each test sequence to the genomic peak set for assessment and applied the findDAR() function with test.method = "exactTest" in SnapATAC version 1.0.0 (*Fang et al., 2021*). The log2FoldDifference was determined for (i) the PV+ cluster relative to all PV- cells using cluster. neg = "random", (ii) the PV+ cluster relative to closely related cells using cluster.neg = "knn", (iii) the PV+ cluster relative to the pool of excitatory neuron clusters, (iv) the PV+ cluster relative to the VIP+ cluster, and (v) the PV+ cluster relative to the SST+ cluster (*Figure 1—figure supplement 5*).

Euarchontoglires PhyloP scores were extracted for all bases within each PV+ enhancer candidate using the UCSC Table Browser (phyloP60wayEuarchontoGlires track for the Grcm38/mm10 genome, accessed March 2021) (*Kuhn et al., 2013*). Regions were mapped from mouse (mm10) to human (hg38) using UCSC LiftOver, requiring a minimum ratio of bases that must remap of 0.1. All regions were mappable between species. Finally, we assessed overlapping human PV+ neuron OCRs from motor cortex snATAC-seq (*Bakken et al., 2020*) using bedtools intersect (*Quinlan and Hall, 2010*). Any peak overlap of at least 1 bp was recorded as an overlapping peak.

## Evaluation of SC1 and SC2 ATAC-seq

PCA was performed using plotPCA() on the DESeqDataSet object with variance stabilizing transformation in DESeq2 version 1.26.0 (*Love et al., 2014*). Using the DESeq2 models described above for cell groups, we extracted OCR statistics for particular cell group comparisons by using the results contrasts. Correlations between log2FoldDifferences for PV+ cSNAIL vs. bulk tissue and log2FoldDifferences for SNAIL probes vs. bulk tissue were assessed using the R function cor.test() with both 'Spearman' and 'Pearson' methods. Genome browser tracks were visualized in the mm10 genome using IGV (*Robinson et al., 2011*) and track heights were normalized between samples of the same experimental ATAC-seq method (cSNAIL, SNAIL, bulk tissue, or single nucleus). Comparisons to snATAC-seq cluster markers (*Figure 3d*, *Figure 4—figure supplement 2*) represent the percentage of cSNAIL/SNAIL ATAC-seq OCRs enriched relative to bulk (padj<0.05 and log2FoldDifference > 0.5) that overlap snATAC-seq cluster markers. snATAC-seq cluster markers were defined as enriched OCRs for that cluster relative to its k-nearest neighbors (padj<0.01 and log2FoldDifference > 1) that were not enriched OCRs for any other cluster. The significance of the enrichments was assessed using the hypergeometric test with the phyper() function in R, setting lower.tail = FALSE. Enrichments for cluster-specific OCRs were assessed using a background of all snATAC-seq OCRs (N = 415,813), and p-values were corrected for 84 tests with Bonferroni correction.

## Assessment of PV+ neuron OCRs in different brain regions

We assessed PV+ neuron cSNAIL ATAC-seq samples from the cortex, striatum, and GPe tissue of healthy control mice from *Lawler et al., 2020* (one male, one female) for differential open chromatin using DESeq2 as described above. We evaluated OCRs that were preferentially open in one brain region relative to each of the other brain regions (padj<0.01 and log2FoldDifference > 1) for sequence motif and pathway enrichments. We computed motif enrichments for tissue-specific PV+ OCRs using AME version 5.3.3 (*McLeay et al., 2010*) against a background of PV+ OCRs from all three tissues. Similarly, we computed pathway enrichments using GREAT version 4.0.4 (*McLean et al., 2010*) for tissue-specific PV+ OCRs relative to a background of PV+ OCRs from all three tissues.

## Model interpretation

We used GkmExplain (*Shrikumar et al., 2019*) to calculate actual and hypothetical importance scores per base for each of 11 SVMs among 1755 true-positive PV+-specific OCR sequences that also scored PV+-specific across all SVMs. First, sequences were one-hot encoded. The importance scores were normalized based on the hypothetical importance scores of all possibilities per base, so that a base position decreased in importance if there were other nucleotide possibilities that produced similar scores. We identified sequence motifs with high contributions to PV+ scores for each SVM separately using TF-MoDISco version 0.4.2.3 (*Shrikumar et al., 2018*) with options chosen to align with final SVM parameters: sliding_window_size = 7, flank_size = 3, min_seqlets_per_task = 3000, trim_to_window_size = 7, initial_flank_to_add = 3, final_flank_to_add = 4, kmer_len = 7, num_gaps = 1, and num_mismatches = 1. The resulting sequence patterns, representing motifs generated from seqlet clusters, were trimmed to the 13 central bases and patterns with support from more than 100 seqlets were used in downstream analysis. The position weight matrices (PWMs) of these patterns were associated with known motifs in the Human and Mouse HOCOMOCO v11 FULL database using Tomtom (*Gupta et al., 2007*) with the Pearson correlation coefficient motif comparison function (*Figure 5—source data 1*). Motifs from all models were clustered based on PWM similarity using STAMP (*Mahony and Benos, 2007*); STAMP operations were performed after trimming motif edges with information content less than 0.4, using ungapped Smith–Waterman alignment, the iterative refinement multiple alignment strategy, Pearson correlation coefficient comparison metrics, and UPGMA tree construction. Finally, individual instances of motif sites were mapped in SC1 and SC2 sequences using FIMO with default parameters (*Grant et al., 2011*).

For in silico mutagenesis of high-importance motifs, we first ranked individual nucleotides based on their GkmExplain importance scores. We used nucleotides in the top 5% of importance scores to estimate motif site boundaries within SC1 and SC2, and we identified the top two important motif sites in each sequence, which were 6–8 bp each. We randomly scrambled motifs and then input the partially scrambled sequences into Tomtom (*Gupta et al., 2007*) to ensure that no other known motif site was introduced. We scored the partially scrambled sequences for each SVM using gkmpredict (*Lee, 2016*) in the same way that we did for other sequences.

---

# Additional information

### Competing interests

Alyssa J Lawler, Easwaran Ramamurthy, Andreas R Pfenning: Inventor on US Patent Application 62/921,452, "Specific nuclear-anchored independent labeling system". The other authors declare that no competing interests exist.

### Funding

| Funder | Grant reference number | Author |
| --- | --- | --- |
| National Institutes of Health | UG3-MH-120094 | William R Stauffer |
| National Institutes of Health | 1DP1DA046585 | Andreas R Pfenning |

| Funder | Grant reference number | Author |
| --- | --- | --- |
| National Science Foundation | DGE1745016 | Alyssa J Lawler |
| National Institute on Drug Abuse | 1F30DA053020 | BaDoi N Phan |
| National Institute of Mental Health | MH096985 | Kenneth N Fish |

The funders had no role in study design, data collection and interpretation, or the decision to submit the work for publication.

## Author contributions

Alyssa J Lawler, Conceptualization, Data curation, Formal analysis, Investigation, Methodology, Visualization, Writing – original draft, Writing – review and editing; Easwaran Ramamurthy, Formal analysis, Methodology, Visualization, Writing – review and editing; Ashley R Brown, Conceptualization, Investigation, Project administration, Resources, Supervision, Writing – review and editing; Naomi Shin, Yeonju Kim, Grant A Fox, Investigation; Noelle Toong, Formal analysis, Investigation; Irene M Kaplow, Morgan Wirthlin, Xiaoyu Zhang, Formal analysis; BaDoi N Phan, Data curation, Formal analysis, Writing – review and editing; Kirsten Wade, Data curation, Investigation, Writing – review and editing; Jing He, Bilge Esin Ozturk, Investigation, Writing – review and editing; Leah C Byrne, William R Stauffer, Kenneth N Fish, Supervision; Andreas R Pfenning, Conceptualization, Funding acquisition, Project administration, Supervision, Writing – review and editing

## Author ORCIDs

Alyssa J Lawler http://orcid.org/0000-0002-2151-5164
Easwaran Ramamurthy http://orcid.org/0000-0002-2439-0600
Ashley R Brown http://orcid.org/0000-0002-3091-3930
Irene M Kaplow http://orcid.org/0000-0002-8924-8269
Morgan Wirthlin http://orcid.org/0000-0001-7967-7070
BaDoi N Phan http://orcid.org/0000-0001-6331-5980
Jing He http://orcid.org/0000-0001-9034-8390
Bilge Esin Ozturk http://orcid.org/0000-0001-5117-077X
Leah C Byrne http://orcid.org/0000-0002-3229-4993
William R Stauffer http://orcid.org/0000-0003-1031-8824
Kenneth N Fish http://orcid.org/0000-0003-1774-3815
Andreas R Pfenning http://orcid.org/0000-0002-3447-9801

## Ethics

This study was performed in strict accordance with the PHS Policy on the Humane Care and Use of Laboratory Animals and the Animal Welfare Act. All animal use and procedures were approved and overseen by the Institutional Animal Care & Use Committee (IACUC) of Carnegie Mellon University (Protocol ID PROTO201600003) or the University of Pittsburgh (Protocol ID 19024431).

## Decision letter and Author response

Decision letter https://doi.org/10.7554/eLife.69571.sa1
Author response https://doi.org/10.7554/eLife.69571.sa2

# Additional files

## Supplementary files
• Supplementary file 1. Sample metadata information.
• Transparent reporting form

## Data availability

Sequencing data have been deposited in GEO under accession code GSE171549.

The following dataset was generated:

| Author(s) | Year | Dataset title | Dataset URL | Database and Identifier |
|---|---|---|---|---|
| Lawler AJ, Ramamurthy E, Brown AR, Shin N, Kim Y, Toong N, Kaplow IM, Wirthlin M, Zhang X, Fox G, Pfenning AR | 2021 | Machine learning sequence prioritization for cell type-specific enhancer design | https://www.ncbi.nlm.nih.gov/geo/query/acc.cgi?acc=GSE171549 | NCBI Gene Expression Omnibus, GSE171549 |

The following previously published datasets were used:

| Author(s) | Year | Dataset title | Dataset URL | Database and Identifier |
|---|---|---|---|---|
| Mo A, Mukamel EA, Davis FP, Luo C, Eddy SR, Ecker JR, Nathans J | 2015 | Epigenomic Signatures of Neuronal Diversity in the Mammalian Brain | https://www.ncbi.nlm.nih.gov/geo/query/acc.cgi?acc=GSE63137 | NCBI Gene Expression Omnibus, GSE63137 |
| Lawler AJ, Brown AR, Bouchard RS, Toong N, Kim Y, Velraj N, Fox G, Kleyman M, Kang B, Gittis AH, Pfenning AR | 2020 | Cell type-specific oxidative stress genomic signatures in the globus pallidus of dopamine depleted mice | https://www.ncbi.nlm.nih.gov/geo/query/acc.cgi?acc=GSE157359 | NCBI Gene Expression Omnibus, GSE157359 |
| Srinivasan C, Phan BN, Lawler AJ, Ramamurthy E, Kleyman M, Brown AR, Kaplow IM, Wirthlin ME, Pfenning AR | 2021 | Addiction-associated genetic variants implicate brain cell type- and region-specific cis-regulatory elements in addiction neurobiology | https://www.ncbi.nlm.nih.gov/geo/query/acc.cgi?acc=GSE161374 | NCBI Gene Expression Omnibus, GSE161374 |

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
