## [Editor Report]

This article describes an exciting new approach for tagging and isolation of unique neuronal subpopulations based on machine learning selection of cell-specific enhancer elements in the genome. The article highlights a specific test case of this technology with neurons expressing Parvalbumin, but this method could be applied to any neuronal or even non-neuronal cell type. The tools and overall approach described here will enable cell tagging in model organisms for which transgenic lines are not commonly available or even expression of other transgenes for control of cell function or genetic perturbation.

---

## [Decision Letter]

**Decision letter after peer review:**

Thank you for submitting your article "Machine learning sequence prioritization for cell type-specific enhancer design" for consideration by *eLife*. Your article has been reviewed by 3 peer reviewers, including Jeremy J Day as Reviewing Editor and Reviewer #1, and the evaluation has been overseen by Naama Barkai as the Senior Editor. The following individual involved in review of your submission has agreed to reveal their identity: Cliff Kentros (Reviewer #3).

Essential revisions:

1) To expand on the overall applicability of the cSNAIL approach, it would be useful to determine whether identified PV-specific sequences using this approach extend to ATAC-seq signal from PV neurons in other species. Showing this would also help to generate confidence that the selected AAV sequences can drive expression in PV neurons in other systems. At a minimum, it would be important to demonstrating the accessibility of the machine-learning identified enhancer sets studied here in publicly available snATAC-seq datasets from other brain regions and in other species. This addition would go a long way to indicating the ability to generalize this approach, which is important for a resource manuscript like this.

2) In many cases, the computational approaches are not fully described and may introduce confusion for a more general readership. It would be useful for descriptions of what each model actually does to be incorporated into the manuscript.

3) Points raised by all reviewers regarding technical and interpretational clarifications should be addressed in a revised manuscript.

*Reviewer #1 (Recommendations for the authors):*

1. The introduction is a bit lengthy, and may be improved by efforts to consolidate the last three paragraphs.

2. It is encouraging that population-derived SVMs and single nucleus-derived SVMs arrive at similar conclusions with respect to selected sequences (Figure 1F). However, if I understand correctly all of this data is from mouse cortex. To extend the applicability of this approach, it would be useful to determine whether identified PV-specific sequences using this approach extend to ATAC-seq signal from PV neurons in other species. Showing this would also help to generate confidence that the selected AAV sequences can drive expression in PV neurons in other systems (thereby significantly expanding the applicability of this tool beyond the mouse).

3. Similar to the above comment, the extension of this approach to other brain regions will largely drive the application of this technology by other groups. While Figure 4 shows data demonstrating that the predictions generated from mouse cortex may hold for mouse striatum and GPe (although not as well as cortex), this claim is only tested using ATAC-seq. This claim could be strengthened by the addition of co-labelling evidence (as in Figure 2 for cortex) that targeted SUN-GFP cells are also PV+.

4. The text identifies Err3 and *Mef2* motifs as being important for the PV-specific activity of SC1 and SC2 sequences. However, without additional experimental evidence demonstrating that loss or mutation of these motifs abolishes the PV-specific expression pattern of these sequences, this conclusion should be moderated.

5. Other reports using open chromatin profiling to identify enhancers for transgene expression in AAV have recently been published (PMID 33789083, 33789096). While these reports are cited here, the Discussion section of this manuscript would benefit from a more systematic comparison of these approaches with the current approach.

*Reviewer #2 (Recommendations for the authors):*

1) Twice the authors compare the PV specificity of their viral vectors to that achieved with the Pvalb-2A-Cre mouse (line 91, line 271), however no data or references are cited to explain what data the authors are comparing from the mouse strain. Published studies using Cre-dependent viruses in adult Pvalb-2A-Cre show excellent specificity for expressing in PV+ neurons, it is when the strain is crossed to other transgenic reporter mice (like the original Ai14 mouse in the Madisen et al., paper) that there is expression outside the PV+ population in the adult brain. This may not reflect Cre expression outside the PV+ population as much as it reflects Pvalb gene activation during development in cells beyond the highly PV+ population of the adult. Therefore, the authors need to clarify what data from the Pvalb-2A-Cre mice to which they are comparing.

2) in Line 275-276, and Figures 2d-e, the authors state that their PV derived enhancer vectors have a 9-fold increase in PV specificity over the negative control. However, the negative control vector has no promoter and thus should really have no expression (there is certainly exceptionally little expression in Figure 2C in the right hand most panel). If the image is truly representative, then it seems that the authors probably have too few expressing cells to say much that is meaningful in a quantitative way about the specificity of that negative control.

3) In line 338 the authors use their motif analysis to suggest a distinction in the *MEF2* family members that may control the differentiation of different inhibitory cell types. However, to my knowledge there is no rigorous experimental identification of distinct MEF2A versus MEF2C versus MEF2D binding sites (if the authors are aware of a validated study this reference would be good to add). The slight variations in the motifs in the databases interrogated here may rather reflect differences in the methods of the groups that deposited the motifs. The authors would benefit the field to think broadly about their interpretation of these variations in motif enrichment and what they might mean.

4) As to the evidence that MEF2C is required for the PV+ interneuron lineage, what the Mayer 2018 paper actually showed is that conditional knockout of MEF2C in PV+ lineage cells (with Dlx6-Cre) led to loss of PV expression in the adult cortex. However, whether this is loss of PV expression, versus failure of PV cells to develop, was not addressed. The authors should adjust their language here to account for that uncertainty. It is also important that the Mayer study settled on MEF2C because it was more highly expressed in PV interneurons compared with other interneurons (esp. SST interneurons). However, MEF2C is highly expressed in many classes of excitatory neurons especially during development but also in the adult brain. Therefore, MEF2C alone cannot explain cell type specific enhancer function except in collaboration with other cell type specific transcription factors.

5) In Line 351 the authors highlight the association of open chromatin near BDNF especially at promoter IV. However, this is somewhat surprising, given that BDNF is really not expressed in interneurons (see PMID 24855953). This example highlights that the authors need to consider the possibility that not every region of differentially open chromatin reflects enhancers – these may instead be regions bound by repressors that silence nearby genes, or architectural factors that affect long-distance regulation of gene expression. The authors could strengthen this aspect of their analysis if they compared their differential OCRs to differential gene expression at least for their major cell types of interest.

*Reviewer #3 (Recommendations for the authors):*

To sum up the public review, it is not that we don't believe that there is added value in the SNAIL approach, we suspect there definitely is, and would honestly like to try it ourselves once it is published. It just needs to be properly and clearly demonstrated. The ideal way to show this would be to analyse the same tissue twice, once via SNAIL and once via any of the competing approaches discussed above, find areas of non-overlap and see which one works best at making specific vectors, but this is admittedly a big ask, which I am not making. However, there needs to be a bit more to demonstrate how useful it might be… another less onerous way one could do so can be seen in Figure 1 suppl 6, which shows the predicted PV specificity from SNAIL (1S6b) versus accessibility alone (1S6c). Note that the two panels are largely similar, but there are particular enhancers which have wildly different predicted specificity than from chromatin accessibility alone. The authors could illustrate how well their approach worked by taking an enhancer with very different values by the two methods (e.g. E14, E12) and seeing which prediction holds true in an AAV: SNAIL vs. raw accessibility.

Specific points are below.

Line 16: "introduce" seems out of place, since cSNAIL was already described in the previous 2020 paper… In general this is a real issue… the paper would be greatly aided by making clearer the *functional* distinctions between the two acronyms, rather than simply say that cSNAIL is Cre-based. The similarity of the acronyms will be confusing to the casual reader who doesn't look up the prior paper, because they are actually quite distinct in application. It is unclear how much of the paper is about the label versus the algorithm (it's almost entirely the latter). This is central to the novelty of the work, so it should be clearer.

Lines 56 to 68: in this description of selection of enhancers one relevant aspect is overlooked, namely that scATAC-seq is a relatively noisy approach to the prediction of active enhancers. A combination of different chromatin marks would likely be equally effective in selecting relevant enhancers. Naturally, this would be harder to accomplish than a simple scATAC-seq. For example the work by Ernst and Kellis is very useful here (for example PMID: 29120462).

Discovery of functional elements does not require necessarily any modeling, the addition of K27Ac marks alone for example improves prediction greatly in the case of active promoters and enhancers.

Figure 1 – supplement 2 and lines 112-114: even though this presentation is convincing in showing the predictive capabilities of machine learning, it does not put the empirically tested promoters into context of all predicted regulatory elements. I would like to see in the text an inclusion of which rank the cell class specific promoters reach (i.e. "the SVM model ranked the Gfap promoter as xx out of xxx astrocyte specific OCRs").

Furthermore, in supplement 2b, I would like to see the inclusion of all enhancers/promoters as gray, points. This will put the verified promoters into context.

Line 130 and other instances: please use "PV+" rather than "PV", in the context of "PV-", only using "PV" for "PV+" is confusing. In the same vein "PV-" may work better to prevent confusion with "PV-specific" in line 132.

Line 133: how many OCRs were present on the merged reproducible list? This information would help to put into context the differentally accessible regions in line 140.

Lines 141 to 153: it is unclear what ROC does for a naïve reader. Please include some more information what actually happens in this model. Furthermore, with OCRs of 500bp and motifs typically 6-20bp (possibly several), that would leave many basepairs of noise. Are the OCRs first screened for relevant motifs, or is the ROC applied directly the full OCRs?

How well does the CNN models perform on the more broad peaks of ATAC-seq compared to the data they were originally designed for with a much higher basepair resolution (ie. CHiP-NEXUS)?

Line 160: how many peaks were present in the merged reproducible list of peaks?

Line 165 to 174: it is unclear what the CNN does for a naïve reader. Please add a few sentences of explanation. Additionally, the true/false positive rate graphs are presented without further explanation.

Lines 221 to 225: Rather than picking out 3 values that correlate with the hypothesis, please report the full data.

In other words, please include a full list/table of enhancers E1-E34 with scores from both models, as well as values of specificity. Then ideally include two graphs with on the X-axis specificity and on the Y axis the model score, for a selection of the contrasts. I requests this because in many cases in figure 1 supplementary 4 and 5 the correlation seems to be driven primarily by E4, E22 and E29.

Could you also provide the correlation between predicted activity and specificity without E4, E22 and E29? This should hold up even without these most obvious examples.

Figure 1 supplement 5a: according to the predicted activity E10, E7 and E9 are particularly specific for PV+ cells compared to VIP+ cells. Similarly E14 should be particularly specific for PV+ cells compared to SST+ cells. This predicted specify is not apparent from the accessibility (supplement 5b) and the general, empirically determined specificity is not particularly high. But, based on the models, these particular enhancers should display specificity for PV+ cells over VIP+ cells. This hypothesis is easily tested by injecting viral vectors with these particular enhancers and counting the transgene expression in PV+ cells compared to VIP+/SST+ cells.

This hypothesis is particularly interesting because the tested enhancers are fully in line with the accessibility predictor of specificity, whereas E10, E7 and E9 predict something different than accessibility.

Figure 1 supplement 5b: It appears the correlation between specificity and accessibility is stronger than the correlation between specificity and predicted activity. Please provide the Pearson correlation of specificity and accessibility also and comment on the difference between the two correlations.

Line 241 to 250: it appears E14, and in lesser extend E11 and E2, have a similarly high specificity. Do these enhancers contain motifs too? For that matter, all other enhancers?

Line 258: what is the motivation for picking SC2 over all other candidates in the 90th percentile?

Figure 2 A-B: It appears the snATAC signal for SC1 and SC2 alone would be very strong in pointing these enhancers out as PV specific enhancers. Based on only Accessibility, how high would they rank? In other words, could you make a list with all enhancers sorted based on log2 fold accessibility difference, and provide the percentile ranks of these enhancers based on this compared to the model predicted ranks?

Figure 2C-E and lines 266 to 276: which region was investigated? Were the same cortical regions investigated to establish the percentages? Some cortical regions are naturally more abundant with PV+ cells. Please include more details on this analysis.

If this is done right: very strong, excellent to see this working! Absolutely convincing SC1 and SC2 drive PV specific expression.

Line 297 to 327: Strong, convincing evidence that SC1 and SC2 are indeed PV specific. This is not only interesting for those researching PV cells, but also an indication that the selection of these enhancers based on the in silico models was successful.

Line 360: it would be interesting to see a GPe or cortical PV+ neuron specific enhancer in a subsequent publication!

Line 444: most interesting, a good lead into functional understanding of enhancers-TF interaction in particular cell types in the brain.

Line 449: This sentence seems a bit of an overstatement. As I understand, first OCRs are selected on differential activity. Meaning a requirement of scATAC data with defined and annontated clusters. For this statement to be true, the models need to be run on all peaks, rather than pre-selected regions with differential activity. Perhaps I misunderstood and the models were actually run on the merged lists of peaks, in that case disregard this comment.

Line 459: I'm not sure which data this is based on. I don't think a direct comparison was made between the average score in Figure 1.sup5a and b. I would like to see this explicitly state in the text, along with correlation plots for specificity vs. predicted activity and specificity vs. accessibility, with pearson correlations.

Line 491: Could you speculate on the possibility to find or generate regulatory elements specific for cell types in these regions. So, instead of generalization, specification.

Lines 493 to 501: Another limitation, is that when supplemented in mature neurons, the viral vector will not undergo the same developmental, epigenomic modifications. This may result in different levels of expression. At least, in our hands we found discrepancies in expression between transgenically provided genes and virally provided ones. There does not seem to be much literature on this topic though, so a discussion may be beyond the scope of this paper.

---

## [Author Response]

Essential revisions:1) To expand on the overall applicability of the cSNAIL approach, it would be useful to determine whether identified PV-specific sequences using this approach extend to ATAC-seq signal from PV neurons in other species. Showing this would also help to generate confidence that the selected AAV sequences can drive expression in PV neurons in other systems. At a minimum, it would be important to demonstrating the accessibility of the machine-learning identified enhancer sets studied here in publicly available snATAC-seq datasets from other brain regions and in other species. This addition would go a long way to indicating the ability to generalize this approach, which is important for a resource manuscript like this.

We agree that demonstrating the generalizability of the approach is of particular importance. We made several substantial computational and experimental advances to address these concerns:

1. We compared the predictions of machine learning models trained on mouse cortex, mouse striatum, and mouse GPe. High concordance between PV+ neuron-specific predictions in the cortex and striatum suggest generalizability between brain regions where PV+ neurons are similar.

2. We trained new machine learning models using human single nucleus open chromatin data (Corces et al., 2020). The success of these models shows the applicability of the SNAIL framework for source data and applications beyond the mouse brain.

3. We demonstrate that PV+ machine learning models trained from human data show substantial correlation to those trained from mouse data. This shows cross-species conservation in the regulatory sequence features that discriminate PV+ neuron-specific enhancers. It also suggests that many machine learning-designed enhancer AAVs chosen from the mouse genome are likely to be transferable to primates and other species. In fact, one of our enhancer candidates SC1 scored as one of the highest predicted PV+ neuron-specific sequences for both mouse and human.

4. We tested one of our SNAIL PV+ AAVs in the macaque cortex where it exhibited similar selectivity for PV+ neurons. This validates the wide utility of the SC1 SNAIL virus and underscores the generalizability of the SNAIL system to other contexts and species.

These results of items 2 – 4 above are described in a new section in the text *“Cross-species analyses with SNAIL”* and reference a new main figure, Figure 6.

2) In many cases, the computational approaches are not fully described and may introduce confusion for a more general readership. It would be useful for descriptions of what each model actually does to be incorporated into the manuscript.

Based on the broad audience of *eLife*, we have added more background and detail about the computational approaches, with special attention to the machine learning modeling. Some examples include:

“These models take 500 bp candidate enhancer DNA sequence strings as input and they output, for each sequence, the cell type in which that candidate enhancer is active and an associated score. The similarity between sequences is determined based on gapped k-mer count vectors, i.e. the number of occurrences of all short subsequences of length k, tolerating some gaps or mismatches, as implemented by LS-GKM (Ghandi et al., 2014; Lee 2016).Where there are sufficient reliable sequence features associated with differential enhancer activation between two cell types, the model should learn these principles during the training phase and then be able to apply these principles to determine the cell type-specific activities of new sequences.”

“CNNs are a type of artificial neural network defined by multiple convolutional layers. The CNNs were trained to take in 1000 bp DNA sequence strings with different accessibility between two cell types, automatically extract predictive sequence features, and output a cell class probability between 0 and 1 (see methods for details). Compared with SVMs, CNNs are better-equipped to learn higher-order interactions between sequence features due to CNNs’ capacity for flexible feature representation and automated feature selection (Cun et al., 1989).”

“To assess model performance, we used standard classifier metrics, the area under the receiver operator curve (auROC) and the area under the precision-recall curve (auPRC). These scores quantify model performance by comparing the predicted class to the actual class, where a randomly guessing binary classifier would have an auROC score of 0.5 and an auPRC score equal to the fraction of actual positives in the data, and a perfect classifier would have a maximum auROC score of 1.0 and auPRC score of 1.0.”

Reviewer #1 (Recommendations for the authors):1. The introduction is a bit lengthy, and may be improved by efforts to consolidate the last three paragraphs.

We consolidated the introduction where possible, with particular attention to the last three paragraphs. In the second to last paragraph, we made the description of machine learning more concise.

“The nucleotide sequence code that links transcription factor binding sites and other DNA features to enhancer activity is underutilized in AAV enhancer design, perhaps due to its complexity (Jindal and Farley 2021). We reasoned that machine learning classifiers could be leveraged to identify the most characteristic and specific enhancer sequence patterns for a cell population, enabling efficient prioritization of sequences that are likely to drive selective expression. Convolutional neural networks (CNNs) (Cun et al., 1989) and support vector machines (SVMs), for example, have achieved state-of-the-art performance on predicting enhancer activity from sequence (Chen, Fish, and Capra 2018; Kaplow et al., 2021; Kelley 2020).”

2. It is encouraging that population-derived SVMs and single nucleus-derived SVMs arrive at similar conclusions with respect to selected sequences (Figure 1F). However, if I understand correctly all of this data is from mouse cortex. To extend the applicability of this approach, it would be useful to determine whether identified PV-specific sequences using this approach extend to ATAC-seq signal from PV neurons in other species. Showing this would also help to generate confidence that the selected AAV sequences can drive expression in PV neurons in other systems (thereby significantly expanding the applicability of this tool beyond the mouse).

We generated new human-based SVMs using single nucleus open chromatin data from human cortical regions (Corces et al., 2020) to parallel the mouse SVMs. Mouse SVM scores and human SVM scores were well correlated (Pearson R=0.77) for PV+ neuron OCRs. We incorporated these additional models into the manuscript in the new Results section *“Cross-species analyses with SNAIL”* line 455 and new Figure 6. We also demonstrated selectivity of the SC1 SNAIL probe for PV+ neurons in the macaque cortex (Figure 6c).

3. Similar to the above comment, the extension of this approach to other brain regions will largely drive the application of this technology by other groups. While Figure 4 shows data demonstrating that the predictions generated from mouse cortex may hold for mouse striatum and GPe (although not as well as cortex), this claim is only tested using ATAC-seq. This claim could be strengthened by the addition of co-labelling evidence (as in Figure 2 for cortex) that targeted SUN-GFP cells are also PV+.

Unfortunately, we did not collect imaging data from the mouse striatum and GPe. We have tempered the language to note that further experiments are necessary to confirm PV+ neuron labeling in additional brain regions. e.g.

“From these interpretations, SC1 and SC2 SNAIL viruses seem to generalize to the striatum and GPe, though they may not be as robust as they are within the cortical context. Additional experimental evidence may be necessary to confirm the appropriateness of PV+ SNAIL viruses for certain applications outside the cortex.”

We also added an additional broader comparison of the machine learning models from different brain regions, which shows high agreement between cortical models and striatal models, and a weak relationship between cortical models and GPe models. This result is biologically intuitive, GPe PV+ neurons have major distinguishing phenotypes, including projections to other brain regions (Hernández et al., 2015; Saunders, Huang, and Sabatini 2016).

“We assessed the correlation between the score outputs of the cortex PV+ vs. PV- SVM with the striatum PV+ vs. PV- or GPe PV+ vs. PV- SVMs for our set of experimentally identified mouse cortical PV+ neuron enhancers sequences. Enhancer scores were well correlated between the cortex and striatum models (pearson = 0.73, spearman = 0.72), indicating shared regulatory sequence determinants between PV+ neurons in the cortex and striatum. There was a low correlation between enhancer scores of the cortex and GPe models (pearson = 0.25, spearman = 0.24), indicating differences in the learned PV+ regulatory sequence properties in these regions.”

Overall, we believe that the ATAC-seq data from multiple brain regions, as well as the addition of the cross-species analyses, demonstrate high utility for PV SNAIL technologies.

4. The text identifies Err3 and Mef2 motifs as being important for the PV-specific activity of SC1 and SC2 sequences. However, without additional experimental evidence demonstrating that loss or mutation of these motifs abolishes the PV-specific expression pattern of these sequences, this conclusion should be moderated.

We agree that further experimentation is needed to assess this claim, and we have moderated the language in the manuscript. We amended the section title to be “Err3 and *Mef2* motifs are important for the PV+ neuron-specific activity score of SC1 and SC2 sequences” to direct our interpretation to the model score and not necessarily to functional impact. To add to this discussion, we have added predictions for the contributions of these motifs using in silico mutagenesis (Figure 5d). We added related text and moderated conclusions in lines 450 – 453:

“Disruption of Err3 and *Mef2* motifs within SC1 or SC2 resulted in sharply decreased the prediction of PV+ specificity according to the scores across the SVMs (Figure 5d). While these impacts are untested in vivo, these analyses provide an intuition for potential nucleotide contributions to PV+ neuron-specific enhancer sequence function in SC1 and SC2.”

5. Other reports using open chromatin profiling to identify enhancers for transgene expression in AAV have recently been published (PMID 33789083, 33789096). While these reports are cited here, the Discussion section of this manuscript would benefit from a more systematic comparison of these approaches with the current approach.

We view SNAIL as a novel addition to existing large screen approaches which could make them more efficient and more accessible for boutique applications. We added to the second paragraph of the discussion to better explain SNAIL in the context of the field:

“Thus far, success in cell type-specific AAV creation has been limited to intensive screens of dozens of individual AAV enhancer candidates in which most do not produce highly selective expression (Vormstein-Schneider et al., 2020; Mich et al., 2021; Graybuck et al., 2021). In SNAIL, our framework for cell type-specific AAV engineering, we incorporate machine learning classifiers as an additional filter for improved enhancer pre-selection in order to mitigate the experimental burden. On a set of externally tested PV+ enhancer-driven AAVs (Vormstein-Schneider et al., 2020), the average PV+ specificity score across our classifiers was more predictive of PV+ specific AAV expression than the log2 fold difference of snATAC-seq signal, sequence conservation, or accessibility conservation at these loci. With the SNAIL framework, we identified and validated two novel enhancers that drive targeted expression in PV+ neurons in the mouse cortex. While these do not represent enough trials to establish a new conversion rate from cell type-specific OCRs to cell type-specific AAVs, we were encouraged by the immediate success of the first probes we selected. We believe that the incorporation of differential sequence property analyses will continue to improve the throughput of targeted AAV development in new contexts and make cell type-specific enhancer tool development accessible for more researchers.”

In later sections of the discussion, we go on to emphasize other unique advantages of SNAIL over other approaches, including the interpretability of sequence contributions (beginning in line 503) and the isolatable reporter Sun1GFP (beginning in line 521).

Reviewer #2 (Recommendations for the authors):1) Twice the authors compare the PV specificity of their viral vectors to that achieved with the Pvalb-2A-Cre mouse (line 91, line 271), however no data or references are cited to explain what data the authors are comparing from the mouse strain. Published studies using Cre-dependent viruses in adult Pvalb-2A-Cre show excellent specificity for expressing in PV+ neurons, it is when the strain is crossed to other transgenic reporter mice (like the original Ai14 mouse in the Madisen et al., paper) that there is expression outside the PV+ population in the adult brain. This may not reflect Cre expression outside the PV+ population as much as it reflects Pvalb gene activation during development in cells beyond the highly PV+ population of the adult. Therefore, the authors need to clarify what data from the Pvalb-2A-Cre mice to which they are comparing.

These comparisons reference data collected alongside SC1 and SC2 and available in this paper. For imaging, we compared to PValb-2A-Cre/Ai14 fluorescent expression (~50% specificity to Pvalb+ cells in our images). We have amended the text to be more explicit about this, and to direct readers to the appropriate source data:

“This was an 11-fold enrichment in precision over the Ef1a promoter and notably, an almost 2-fold enrichment over Cre reporter labeling in Pvalb-2A-Cre/Ai14 double transgenic mice (Figure 2c-e, Figure 2-Source Data 2).”

For the affinity purification ATAC-seq data, we compared to single transgenic PValb-2A-Cre labeling using cSNAIL. Our claim that SNAIL probes are more specific to PV+ interneurons than the PValb-2A-Cre strain references the experiment comparing all Sun1GFP-purified populations with snATAC-seq (Figure 3d).

“We also note that cSNAIL PV+ ATAC-seq had an additional 8% overlap with excitatory cluster L5 PT markers (p = 2.5 x 10-45), possibly reflective of Pvalb-2A-Cre line labeling in layer 5 Parvalbumin-expressing excitatory neurons (Tanahira et al., 2009; Jinno and Kosaka 2004; Roccaro-Waldmeyer et al., 2018). These OCRs were absent in SC1- and SC2-isolated cells. In fact, SC1 and SC2 had no enrichment for cluster-specific OCRs of any cluster other than PV+ (≤ 2% overlap, p > 0.1), including the closely related SST+ neuron population. This suggests that SC1 and SC2 SNAIL probes target a stricter subset of the cells than the Pvalb-2A-Cre mouse strain, likely restricted to PV+ inhibitory interneurons.”

2) in Line 275-276, and Figures 2d-e, the authors state that their PV derived enhancer vectors have a 9-fold increase in PV specificity over the negative control. However, the negative control vector has no promoter and thus should really have no expression (there is certainly exceptionally little expression in Figure 2C in the right hand most panel). If the image is truly representative, then it seems that the authors probably have too few expressing cells to say much that is meaningful in a quantitative way about the specificity of that negative control.

There were 154 weakly expressing gfp+ cells total across the N.C. images, compared to thousands of gfp+ for SC1 and SC2 (see Figure 2-Source Data 2). We have removed the statement about the 9-fold increase over N.C. because there was a difference between titers. Instead, we use the N.C. data to determine that there is no PV+ bias in background expression. We have changed the text to reflect these differences:

“The negative control virus was injected at as high of a concentration as possible (14x the concentration of SC1, SC2, and Ef1a injections) to detect any biases in spurious background expression. At this titer, a number of cells exhibited GFP expression (Figure 2-Source Data 2), but these did not appear biased toward PV+ neurons (Figure 2d).”

3) In line 338 the authors use their motif analysis to suggest a distinction in the MEF2 family members that may control the differentiation of different inhibitory cell types. However, to my knowledge there is no rigorous experimental identification of distinct MEF2A versus MEF2C versus MEF2D binding sites (if the authors are aware of a validated study this reference would be good to add). The slight variations in the motifs in the databases interrogated here may rather reflect differences in the methods of the groups that deposited the motifs. The authors would benefit the field to think broadly about their interpretation of these variations in motif enrichment and what they might mean.

We have modified the paragraph beginning on line 429 to remove the claims about different *Mef2* motifs. The utility and interpretations are not heavily dependent on the specificities of the *Mef2* family members.

4) As to the evidence that MEF2C is required for the PV+ interneuron lineage, what the Mayer 2018 paper actually showed is that conditional knockout of MEF2C in PV+ lineage cells (with Dlx6-Cre) led to loss of PV expression in the adult cortex. However, whether this is loss of PV expression, versus failure of PV cells to develop, was not addressed. The authors should adjust their language here to account for that uncertainty. It is also important that the Mayer study settled on MEF2C because it was more highly expressed in PV interneurons compared with other interneurons (esp. SST interneurons). However, MEF2C is highly expressed in many classes of excitatory neurons especially during development but also in the adult brain. Therefore, MEF2C alone cannot explain cell type specific enhancer function except in collaboration with other cell type specific transcription factors.

Thank you for the suggestion. We completely agree that MEF2C alone cannot explain cell type-specificity. In fact, the strong performance of our computational models is almost certainly due to their ability to identify how complex combinations of transcription factor binding sites and other sequence signatures influence PV+ neuron-specific enhancer activity. We have added text to the discussion that clarifies the role we propose MEF2C plays in this system.

“An additional advantage of incorporating classifiers for cell type-specific enhancer selection is increased interpretability of the factors that govern success. The sequence patterns learned by PV+ models reflected known PV+ neuron biology. Common motifs contributing to successful PV+ probe enhancers included Err3, *Mef2*, and Lhx6, important for the specification and maintenance of the cortical PV+ interneuron lineage (Zhao et al., 2008; Mayer et al., 2018; Liodis et al., 2007). It is interesting to note the varying expression patterns of these transcription factors. In the cortex, Err3 expression is mainly restricted to PV+ neurons and Lhx6 is restricted to MGE interneurons, but *Mef2* transcripts are widely expressed in many inhibitory and excitatory neurons. It is likely combinatorial interactions between abundant and cell type-specific transcription factor binding events that specify PV+ neuron enhancer activation. The SNAIL framework provides an opportunity to meaningfully leverage these complex sequence codes.”

5) In Line 351 the authors highlight the association of open chromatin near BDNF especially at promoter IV. However, this is somewhat surprising, given that BDNF is really not expressed in interneurons (see PMID 24855953). This example highlights that the authors need to consider the possibility that not every region of differentially open chromatin reflects enhancers – these may instead be regions bound by repressors that silence nearby genes, or architectural factors that affect long-distance regulation of gene expression. The authors could strengthen this aspect of their analysis if they compared their differential OCRs to differential gene expression at least for their major cell types of interest.

We have modified the BDNF OCR discussion and included speculation on this discrepancy.

“The set of PV+ OCRs enriched in cortical PV+ neurons included 10 regions associated with the Bdnf gene (Ensembl Genes; FDR Q = 0.0035). Bdnf is generally expressed in excitatory forebrain neurons but not PV+ interneurons. The presence of PV+ neuron OCRs near Bdnf could represent genomic regions with non-enhancer functions, enhancers that regulate another gene, the binding regions of repressive TFs, or trace contamination from excitatory populations.”

Reviewer #3 (Recommendations for the authors):To sum up the public review, it is not that we don't believe that there is added value in the SNAIL approach, we suspect there definitely is, and would honestly like to try it ourselves once it is published. It just needs to be properly and clearly demonstrated. The ideal way to show this would be to analyse the same tissue twice, once via SNAIL and once via any of the competing approaches discussed above, find areas of non-overlap and see which one works best at making specific vectors, but this is admittedly a big ask, which I am not making. However, there needs to be a bit more to demonstrate how useful it might be… another less onerous way one could do so can be seen in Figure 1 suppl 6, which shows the predicted PV specificity from SNAIL (1S6b) versus accessibility alone (1S6c). Note that the two panels are largely similar, but there are particular enhancers which have wildly different predicted specificity than from chromatin accessibility alone. The authors could illustrate how well their approach worked by taking an enhancer with very different values by the two methods (e.g. E14, E12) and seeing which prediction holds true in an AAV: SNAIL vs. raw accessibility.Specific points are below.Line 16: "introduce" seems out of place, since cSNAIL was already described in the previous 2020 paper… In general this is a real issue… the paper would be greatly aided by making clearer the functional distinctions between the two acronyms, rather than simply say that cSNAIL is Cre-based. The similarity of the acronyms will be confusing to the casual reader who doesn't look up the prior paper, because they are actually quite distinct in application. It is unclear how much of the paper is about the label versus the algorithm (it's almost entirely the latter). This is central to the novelty of the work, so it should be clearer.

We have made attempts throughout the manuscript to clarify. Specifically, we note the similarity and difference between cSNAIL and SNAIL at first mention, line 90:

“Similar to our previously described Cre-activated AAV technology cSNAIL (Lawler et al., 2020), SNAIL probes have the unique advantage of an affinity purification-compatible fluorescent reporter that can be used to isolate rare cell types (Mo et al., 2015; Deal and Henik_off_ 2010; Lawler et al., 2020). Unlike cSNAIL, which relies on Cre activation, SNAIL probes are stand-alone AAVs driven by cell type-specific enhancer sequences selected through machine learning models. Thus, SNAIL probes may be used in a wide variety of systems including wild type mice, primates, and other species.”

We also changed our language to refer to SNAIL throughout the text as a “framework” for machine learning-assisted cell type-specific sequence design. For example:

“Here, we present a framework for machine learning-assisted engineering of cell type-specific AAVs, which we refer to as Specific Nuclear Anchored Independent Labeling (SNAIL).”

“In SNAIL, our framework for cell type-specific AAV engineering, we incorporate machine learning classifiers as an additional filter for improved enhancer pre-selection in order to mitigate the experimental burden.”

“With the SNAIL framework, we identified and validated two novel enhancers that drive targeted expression in PV+ neurons in the mouse cortex.”

Thank you, it is corrected.

Lines 56 to 68: in this description of selection of enhancers one relevant aspect is overlooked, namely that scATAC-seq is a relatively noisy approach to the prediction of active enhancers. A combination of different chromatin marks would likely be equally effective in selecting relevant enhancers. Naturally, this would be harder to accomplish than a simple scATAC-seq. For example the work by Ernst and Kellis is very useful here (for example PMID: 29120462).Discovery of functional elements does not require necessarily any modeling, the addition of K27Ac marks alone for example improves prediction greatly in the case of active promoters and enhancers.

We have added to that paragraph to note that ATAC-seq is an imperfect proxy for enhancer activity.

“ATAC-seq (Buenrostro et al., 2013) is an advantageous technique for defining potential cell type-specific enhancer regions because of its high nucleotide resolution and its compatibility with small cell populations and even with single cell technologies (Buenrostro, Wu, Litzenburger, et al., 2015; Cusanovich et al., 2015). Though convenient, chromatin accessibility is a noisy approximation for enhancer activity and differentially accessible sequence elements often fail to produce selective expression in vivo.”

We agree that the addition of other epigenetic features, including H3K27ac and H3K4me1 (Ernst and Kellis 2017), could potentially improve the predictions. These, and other potential features, are not included in our approach for two reasons. First, these histone modifications tend to be found across broader regions than ATAC-Seq peaks (>1,000bp) (Roadmap Epigenomics Consortium [2015] 2015). In our experience, this can make it more difficult to learn important local sequence features. Second, the availability of this type of data for specific cell types is rare, because of the technical difficulty of ChIP-seq in tiny populations of cells. Overall, our approach helps isolate true cell type-specific sequence characteristics while only requiring convenient ATAC-seq data.

Figure 1 – supplement 2 and lines 112-114: even though this presentation is convincing in showing the predictive capabilities of machine learning, it does not put the empirically tested promoters into context of all predicted regulatory elements. I would like to see in the text an inclusion of which rank the cell class specific promoters reach (i.e. "the SVM model ranked the Gfap promoter as xx out of xxx astrocyte specific OCRs").Furthermore, in supplement 2b, I would like to see the inclusion of all enhancers/promoters as gray, points. This will put the verified promoters into context.

We have added the rank information as suggested. We also included all test set enhancers as gray points in Figure 1—figure supplement 2b.

“The Gfap promoter sequence, which has a heavy astrocyte bias in vivo, was classified as astrocyte-specific in our neuron vs. astrocyte model (5,580 astrocyte-specific enhancers evaluated; Gfap ranks 3,298/5,580). The same neuron vs. astrocyte model scored the CamkII promoter and Dlx promoter sequences, which are known to have neuron-specific activity, as neuron-specific (14,347 neuron-specific enhancers evaluated; CamkII = rank 13,300/14,347, Dlx = rank 9,736/14,347). Also consistent with empirical expectations, the excitatory vs. inhibitory neuron model predicted the CamkII sequence to have an excitatory neuron preference and the Dlx sequence to have an inhibitory neuron preference (15,391 excitatory neuron-specific enhancers evaluated; CamkII = rank 11,541. 4,608 inhibitory neuron-specific enhancers tested; Dlx = rank 1,142/4,608) (Figure 1—figure supplement 2).”

Line 130 and other instances: please use "PV+" rather than "PV", in the context of "PV-", only using "PV" for "PV+" is confusing. In the same vein "PV-" may work better to prevent confusion with "PV-specific" in line 132.

All instances have been standardized to PV+ and PV-.

Line 133: how many OCRs were present on the merged reproducible list? This information would help to put into context the differentally accessible regions in line 140.

There were 160,450 merged reproducible peaks in this analysis. We added detailed information about the number of total peaks, differential peaks, and positive/negative set ratios for all comparisons into Figure 1-Source Data 1.

Lines 141 to 153: it is unclear what ROC does for a naïve reader. Please include some more information what actually happens in this model. Furthermore, with OCRs of 500bp and motifs typically 6-20bp (possibly several), that would leave many basepairs of noise. Are the OCRs first screened for relevant motifs, or is the ROC applied directly the full OCRs?

We have included a better description of what the models actually do at first mention. We do not screen OCRs for relevant motifs, and the full 500 bp sequence goes into the model and evaluations. A key advantage of the machine learning models is their ability to pick out relevant sequence signal (e.g. motifs) among noise, much better than a person could do manually with this volume of data.

“These models take 500 bp candidate enhancer DNA sequence strings as input and they output, for each sequence, the cell type in which that candidate enhancer is active and an associated score. The similarity between sequences is determined based on gapped k-mer count vectors, i.e. the number of occurrences of all short subsequences of length k, tolerating some gaps or mismatches, as implemented by LS-GKM (Ghandi et al., 2014; D. Lee 2016). Where there are sufficient reliable sequence features associated with differential enhancer activation between two cell types, the model should learn these principles during the training phase and then be able to apply these principles to determine the cell type-specific activities of new sequences.”

ROC is a measure of accuracy of the classifier, i.e., given the 500bp sequence alone, how well can the model correctly classify sequences into the PV+ or PV- category. ROC metrics are applied to the whole 500 bp sequence, where each 500 bp sequence counts as 1 true positive, false positive, true negative, or false negative.

How well does the CNN models perform on the more broad peaks of ATAC-seq compared to the data they were originally designed for with a much higher basepair resolution (ie. CHiP-NEXUS)?

CNNs have been widely adopted for a large variety of data modalities, from computer vision to natural language processing and genomics. Within genomics, they have been applied to wide variety of sequence signals. Some of the early approaches were indeed shorter sequences for transcription factor binding sites (Zeng et al., 2016). However, when predicting broader epigenetic features, including enhancers, 1000bp can be used to train accurate models (Zhou and Troyanskaya 2015). Subsequent models have under even longer 3kb inputs (Kelley 2020). Some recent methods have even used sequences over 100kb as input to predictions (Kelley 2020).

Line 160: how many peaks were present in the merged reproducible list of peaks?

There were 186,016 merged reproducible peaks. I have added all compiled peak information in Figure 1—figure supplement 1.

Line 165 to 174: it is unclear what the CNN does for a naïve reader. Please add a few sentences of explanation. Additionally, the true/false positive rate graphs are presented without further explanation.

We have added clarification on the CNN modeling and the graphs of auROC and auPRC.

“CNNs are a type of artificial neural network defined by multiple convolutional layers. The CNNs were trained to take in 1000 bp DNA sequence strings with different accessibility between two cell types, automatically extract predictive sequence features, and output a cell class probability between 0 and 1 (see methods for details). Compared with SVMs, CNNs are better-equipped to learn higher-order interactions between sequence features due to CNNs’ capacity for flexible feature representation and automated feature selection (Cun et al., 1989).”

“To assess model performance, we used standard classifier metrics, the area under the receiver operator curve (auROC) and the area under the precision-recall curve (auPRC). These scores quantify model performance by comparing the predicted class to the actual class, where a randomly guessing binary classifier would have an auROC score of 0.5 and an auPRC score equal to the fraction of actual positives in the data, and a perfect classifier would have a maximum auROC score of 1.0 and auPRC score of 1.0.”

Lines 221 to 225: Rather than picking out 3 values that correlate with the hypothesis, please report the full data.In other words, please include a full list/table of enhancers E1-E34 with scores from both models, as well as values of specificity. Then ideally include two graphs with on the X-axis specificity and on the Y axis the model score, for a selection of the contrasts. I requests this because in many cases in figure 1 supplementary 4 and 5 the correlation seems to be driven primarily by E4, E22 and E29.Could you also provide the correlation between predicted activity and specificity without E4, E22 and E29? This should hold up even without these most obvious examples.

We have included the full data in a new supplement Figure 1-Source Data 3. Based on the feedback, we have transitioned away from correlation estimates and toward binary analysis. This is because our models are binary classifiers, which predict a class label (+ or -), and the score should be thought of as a measure of confidence of this label, with scores near 0 being less confident, and scores further from 0 more confident. Therefore, we should not necessarily expect strong correlation between model scores and experimental specificity for enhancers with intermediate performance. A regression model would be more appropriate if trying to recapitulate this trend. Instead, as you note, our classifiers perform great on the best and worst examples. This is useful for narrowing a pool of candidate enhancers down to the best candidates. Essentially, the practical goal is to minimize false positives, where a high LFC might lead you to use an enhancer, but it actually has poor specificity in vivo due to sequence composition. We have revised this analysis to group E1 – E34 into either high experimental specificity (>70%) or low experimental specificity (<70%), and plotted the LFC or model scores in Figure 1—figure supplement 5. The results are described in the text:

“On the complete set of enhancers, SVM score was predictive of the PV+ specificity group (p = 0.008), and differential activity log2 fold change had a weak association with PV+ specificity group (p = 0.069) (Figure 1—figure supplement 5c). Much of the log2 fold difference association with group PV+ specificity was driven by enhancer sequences with low log2 fold difference and low PV+ specificity. Undesirably, there were some sequences with high log2 fold difference and low in vivo specificity. Within the subset of enhancers with high log2 fold difference (log2 fold difference > 1), the log2 fold difference was not associated with specificity group (p = 0.601), while SVM score was weakly associated (p = 0.057) (Figure 1—figure supplement 5d). Unlike log2 fold difference scores alone, SVM scores may limit false positive candidates and improve efficient enhancer AAV selection.”

Figure 1 supplement 5a: according to the predicted activity E10, E7 and E9 are particularly specific for PV+ cells compared to VIP+ cells. Similarly E14 should be particularly specific for PV+ cells compared to SST+ cells. This predicted specify is not apparent from the accessibility (supplement 5b) and the general, empirically determined specificity is not particularly high. But, based on the models, these particular enhancers should display specificity for PV+ cells over VIP+ cells. This hypothesis is easily tested by injecting viral vectors with these particular enhancers and counting the transgene expression in PV+ cells compared to VIP+/SST+ cells.This hypothesis is particularly interesting because the tested enhancers are fully in line with the accessibility predictor of specificity, whereas E10, E7 and E9 predict something different than accessibility.

Thank you for the suggestion. It is a great experiment idea, but not within the scope of this project at this time.

Figure 1 supplement 5b: It appears the correlation between specificity and accessibility is stronger than the correlation between specificity and predicted activity. Please provide the Pearson correlation of specificity and accessibility also and comment on the difference between the two correlations.

We appreciate this suggestion. Based on the new models we constructed, we believe this comment no longer applies. See above note on binarized analysis.

Line 241 to 250: it appears E14, and in lesser extend E11 and E2, have a similarly high specificity. Do these enhancers contain motifs too? For that matter, all other enhancers?

Other enhancers do contain PV+ relevant motifs to some extent as well. We have included a new supplementary table, Figure 1-Source Data 4, with this information.

Line 258: what is the motivation for picking SC2 over all other candidates in the 90th percentile?

We used numerous criteria for identifying enhancers that are outlined in the methods (beginning line 746, section “SVM data preparation”). After scoring with all models (all scores available in Figure 2-Source Data 1), we arbitrarily chose one at the 90^th^ percentile to get a better sense of whether only the very top enhancers are likely to drive cell type-specificity. We amended the main text to explain this:

“Among true PV+ neuron-specific enhancer sequences that i) were differential OCRs in PV+ vs. PV-, PV+ vs. EXC, and PV+ vs. VIP+ sorted population data and ii) scored PV+ positive across all SVM evaluations (1,755 sequences), SC1 was the highest predicted sequence candidate, while SC2 was in the 90th percentile (Figure 2b, Figure 2-Source Data 1). We chose these sequences to represent two different confidence levels and evaluate the general potential of the top 10% of machine learning-prioritized PV+ neuron-specific enhancers.”

Figure 2 A-B: It appears the snATAC signal for SC1 and SC2 alone would be very strong in pointing these enhancers out as PV specific enhancers. Based on only Accessibility, how high would they rank? In other words, could you make a list with all enhancers sorted based on log2 fold accessibility difference, and provide the percentile ranks of these enhancers based on this compared to the model predicted ranks?

We have provided the accessibility log2 fold difference from the PV+ vs PV- contrast from our sorted population cSNAIL data in Figure 2—figure supplement 1. Based only on accessibility log2 fold difference, SC1 ranks in the 85^th^ percentile and SC2 ranks in the 99^th^ percentile. We argue that log2 fold difference is informative, but not sufficient for probe selection as it calls many false positives. Our models provide an additional filter for efficient narrowing down of candidates and enable interpretation of candidate probe sequences.

Figure 2C-E and lines 266 to 276: which region was investigated? Were the same cortical regions investigated to establish the percentages? Some cortical regions are naturally more abundant with PV+ cells. Please include more details on this analysis.If this is done right: very strong, excellent to see this working! Absolutely convincing SC1 and SC2 drive PV specific expression.

Yes, these were all consistently imaged in the primary motor cortex (M1), with representation from all layers. I have added a note to the text of this section.

“We quantified images with consistent positioning in the primary motor cortex, with representation from all layers.”

Line 297 to 327: Strong, convincing evidence that SC1 and SC2 are indeed PV specific. This is not only interesting for those researching PV cells, but also an indication that the selection of these enhancers based on the in silico models was successful.Line 360: it would be interesting to see a GPe or cortical PV+ neuron specific enhancer in a subsequent publication!Line 444: most interesting, a good lead into functional understanding of enhancers-TF interaction in particular cell types in the brain.Line 449: This sentence seems a bit of an overstatement. As I understand, first OCRs are selected on differential activity. Meaning a requirement of scATAC data with defined and annontated clusters. For this statement to be true, the models need to be run on all peaks, rather than pre-selected regions with differential activity. Perhaps I misunderstood and the models were actually run on the merged lists of peaks, in that case disregard this comment.

The reviewer’s interpretation is correct. We have amended to qualify the statement.

“Here, we showed that sequence was sufficient to discern the directionality of highly differential OCR activity in different neuron subtypes in most cases.”

Line 459: I'm not sure which data this is based on. I don't think a direct comparison was made between the average score in Figure 1.sup5a and b. I would like to see this explicitly state in the text, along with correlation plots for specificity vs. predicted activity and specificity vs. accessibility, with pearson correlations.

We have now included explicit statements of the binary analysis in the text, which supports this claim.

“On the complete set of enhancers, SVM score was predictive of the PV+ specificity group (p = 0.008), and differential activity log2 fold difference had a weak association with PV+ specificity group (p = 0.069) (Figure 1—figure supplement 5c). Much of the log2 fold difference association with group PV+ specificity was driven by enhancer sequences with low log2 fold difference and low PV+ specificity. Undesirably, there were some sequences with high log2 fold difference and low in vivo specificity. Within the subset of enhancers with high log2 fold difference (log2 fold difference > 1), the log2 fold difference was not associated with specificity group (p = 0.601), while SVM score was weakly associated (p = 0.057) (Figure 1—figure supplement 5d). Unlike log2 fold difference scores alone, SVM scores may limit false positive candidates and improve efficient enhancer AAV selection.”

Line 491: Could you speculate on the possibility to find or generate regulatory elements specific for cell types in these regions. So, instead of generalization, specification.

We have added a sentence to state the possibility of using the SNAIL framework to tag specialized populations such as this.

“It will be interesting to conduct further comparisons of the composition of cell populations that are labeled by these tools in different brain regions and species. We speculate that the SNAIL framework could be strategically employed to design tools for specific subtypes of PV+ neurons or other cell populations that may be brain region or species specific.”

Lines 493 to 501: Another limitation, is that when supplemented in mature neurons, the viral vector will not undergo the same developmental, epigenomic modifications. This may result in different levels of expression. At least, in our hands we found discrepancies in expression between transgenically provided genes and virally provided ones. There does not seem to be much literature on this topic though, so a discussion may be beyond the scope of this paper.

We have added a sentence on this potential limitation.

“Finally, an enhancer’s activity in AAV may differ from the same enhancer’s activity in the host genome due to the surrounding genomic and epigenomic context. Enhancer activity may also fluctuate in response to different conditions because enhancers are dynamic actors in the regulation of gene expression.”

References

Buenrostro, Jason D., Paul G. Giresi, Lisa C. Zaba, Howard Y. Chang, and William J. Greenleaf. 2013. “Transposition of Native Chromatin for Fast and Sensitive Epigenomic Profiling of Open Chromatin, DNA-Binding Proteins and Nucleosome Position.” *Nature Methods* 10 (12): 1213–18.

Buenrostro, Jason D., Beijing Wu, Ulrike M. Litzenburger, Dave Ruff, Michael L. Gonzales, Michael P. Snyder, Howard Y. Chang, and William J. Greenleaf. 2015. “Single-Cell Chromatin Accessibility Reveals Principles of Regulatory Variation.” *Nature* 523 (7561): 486–90.

Chen, Ling, Alexandra E. Fish, and John A. Capra. 2018. “Prediction of Gene Regulatory Enhancers across Species Reveals Evolutionarily Conserved Sequence Properties.” *PLoS Computational Biology* 14 (10): e1006484.

Corces, M. Ryan, Anna Shcherbina, Soumya Kundu, Michael J. Gloudemans, Laure Frésard, Jeffrey M. Granja, Bryan H. Louie, et al. 2020. “Single-Cell Epigenomic Analyses Implicate Candidate Causal Variants at Inherited Risk Loci for Alzheimer’s and Parkinson's Diseases.” *Nature Genetics* 52 (11): 1158–68.

Cun, Y. L., L. D. Jackel, B. Boser, J. S. Denker, H. P. Graf, I. Guyon, D. Henderson, R. E. Howard, and W. Hubbard. 1989. “Handwritten Digit Recognition: Applications of Neural Network Chips and Automatic Learning.” *IEEE Communications Magazine*. https://doi.org/10.1109/35.41400.

Cusanovich, Darren A., Riza Daza, Andrew Adey, Hannah A. Pliner, Lena Christiansen, Kevin L. Gunderson, Frank J. Steemers, Cole Trapnell, and Jay Shendure. 2015. “Multiplex Single Cell Profiling of Chromatin Accessibility by Combinatorial Cellular Indexing.” *Science* 348 (6237): 910–14.

Deal, Roger B., and Steven Henikoff. 2010. “A Simple Method for Gene Expression and Chromatin Profiling of Individual Cell Types within a Tissue.” *Developmental Cell* 18 (6): 1030–40.

Ernst, Jason, and Manolis Kellis. 2017. “Chromatin-State Discovery and Genome Annotation with ChromHMM.” *Nature Protocols* 12 (12): 2478–92.

Ghandi, Mahmoud, Dongwon Lee, Morteza Mohammad-Noori, and Michael A. Beer. 2014. “Enhanced Regulatory Sequence Prediction Using Gapped K-Mer Features.” *PLoS Computational Biology* 10 (7): e1003711.

Graybuck, Lucas T., Tanya L. Daigle, Adriana E. Sedeño-Cortés, Miranda Walker, Brian Kalmbach, Garreck H. Lenz, Elyse Morin, et al. 2021. “Enhancer Viruses for Combinatorial Cell-Subclass-Specific Labeling.” *Neuron*, March. https://doi.org/10.1016/j.neuron.2021.03.011.

Hernández, Vivian M., Daniel J. Hegeman, Qiaoling Cui, Daniel A. Kelver, Michael P. Fiske, Kelly E. Glajch, Jason E. Pitt, Tina Y. Huang, Nicholas J. Justice, and C. Savio Chan. 2015. “Parvalbumin+ Neurons and Npas1+ Neurons Are Distinct Neuron Classes in the Mouse External Globus Pallidus.” *The Journal of Neuroscience: The Official Journal of the Society for Neuroscience* 35 (34): 11830–47.

Jindal, Granton A., and Emma K. Farley. 2021. “Enhancer Grammar in Development, Evolution, and Disease: Dependencies and Interplay.” *Developmental Cell* 56 (5): 575–87.

Jinno, Shozo, and Toshio Kosaka. 2004. “Parvalbumin Is Expressed in Glutamatergic and GABAergic Corticostriatal Pathway in Mice.” *The Journal of Comparative Neurology* 477 (2): 188–201.

Kaplow, Irene M., Daniel E. Schäffer, Morgan E. Wirthlin, Alyssa J. Lawler, Ashley R. Brown, Michael Kleyman, and Andreas R. Pfenning. 2021. “Predicting Lineage-Specific Differences in Open Chromatin across Dozens of Mammalian Genomes.” *bioRxiv*. https://doi.org/10.1101/2020.12.04.410795.

Kelley, David R. 2020. “Cross-Species Regulatory Sequence Activity Prediction.” *PLoS Computational Biology* 16 (7): e1008050.

Lawler, Alyssa J., Ashley R. Brown, Rachel S. Bouchard, Noelle Toong, Yeonju Kim, Nitinram Velraj, Grant Fox, et al. 2020. “Cell Type-Specific Oxidative Stress Genomic Signatures in the Globus Pallidus of Dopamine-Depleted Mice.” *The Journal of Neuroscience* 40 (50): 9772–83.

Lee, Dongwon. 2016. “LS-GKM: A New Gkm-SVM for Large-Scale Datasets.” *Bioinformatics* 32 (14): 2196–98.

Liodis, Petros, Myrto Denaxa, Marirena Grigoriou, Cynthia Akufo-Addo, Yuchio Yanagawa, and Vassilis Pachnis. 2007. “Lhx6 Activity Is Required for the Normal Migration and Specification of Cortical Interneuron Subtypes.” *The Journal of Neuroscience: The Official Journal of the Society for Neuroscience* 27 (12): 3078–89.

Li, Yang Eric, Sebastian Preissl, Xiaomeng Hou, Ziyang Zhang, Kai Zhang, Yunjiang Qiu, Olivier B. Poirion, et al. 2021. “An Atlas of Gene Regulatory Elements in Adult Mouse Cerebrum.” *Nature* 598 (7879): 129–36.

Mayer, Christian, Christoph Hafemeister, Rachel C. Bandler, Robert Machold, Renata Batista Brito, Xavier Jaglin, Kathryn Allaway, Andrew Butler, Gord Fishell, and Rahul Satija. 2018. “Developmental Diversification of Cortical Inhibitory Interneurons.” *Nature* 555 (7697): 457–62.

McLean, Cory Y., Dave Bristor, Michael Hiller, Shoa L. Clarke, Bruce T. Schaar, Craig B. Lowe, Aaron M. Wenger, and Gill Bejerano. 2010. “GREAT Improves Functional Interpretation of Cis-Regulatory Regions.” *Nature Biotechnology* 28 (5): 495–501.

Mich, John K., Lucas T. Graybuck, Erik E. Hess, Joseph T. Mahoney, Yoshiko Kojima, Yi Ding, Saroja Somasundaram, et al. 2021. “Functional Enhancer Elements Drive Subclass-Selective Expression from Mouse to Primate Neocortex.” *Cell Reports* 34 (13): 108754.

Mo, Alisa, Eran A. Mukamel, Fred P. Davis, Chongyuan Luo, Gilbert L. Henry, Serge Picard, Mark A. Urich, et al. 2015. “Epigenomic Signatures of Neuronal Diversity in the Mammalian Brain.” *Neuron* 86 (6): 1369–84.

Roadmap Epigenomics Consortium. (2015) 2015. “Integrative Analysis of 111 Reference Human Epigenomes.” *Nature* 518 (7539): 317–30.

Roccaro-Waldmeyer, Diana M., Franck Girard, Daniele Milani, Elisabetta Vannoni, Laurent Prétôt, David P. Wolfer, and Marco R. Celio. 2018. “Eliminating the VGlut2-Dependent Glutamatergic Transmission of Parvalbumin-Expressing Neurons Leads to Deficits in Locomotion and Vocalization, Decreased Pain Sensitivity, and Increased Dominance.” *Frontiers in Behavioral Neuroscience* 12: 146.

Saunders, Arpiar, Kee Wui Huang, and Bernardo Luis Sabatini. 2016. “Globus Pallidus Externus Neurons Expressing Parvalbumin Interconnect the Subthalamic Nucleus and Striatal Interneurons.” *PloS One* 11 (2): e0149798.

Tanahira, Chiyoko, Shigeyoshi Higo, Keisuke Watanabe, Ryohei Tomioka, Satoe Ebihara, Takeshi Kaneko, and Nobuaki Tamamaki. 2009. “Parvalbumin Neurons in the Forebrain as Revealed by Parvalbumin-Cre Transgenic Mice.” *Neuroscience Research* 63 (3): 213–23.

Vormstein-Schneider, Douglas, Jessica D. Lin, Kenneth A. Pelkey, Ramesh Chittajallu, Baolin Guo, Mario A. Arias-Garcia, Kathryn Allaway, et al. 2020. “Viral Manipulation of Functionally Distinct Interneurons in Mice, Non-Human Primates and Humans.” *Nature Neuroscience*, August. https://doi.org/10.1038/s41593-020-0692-9.

Zeng, Haoyang, Matthew D. Edwards, Ge Liu, and David K. Gifford. 2016. “Convolutional Neural Network Architectures for Predicting DNA–protein Binding.” *Bioinformatics*. https://doi.org/10.1093/bioinformatics/btw255.

Zhao, Yangu, Pierre Flandin, Jason E. Long, Melissa Dela Cuesta, Heiner Westphal, and John L. R. Rubenstein. 2008. “Distinct Molecular Pathways for Development of Telencephalic Interneuron Subtypes Revealed through Analysis of Lhx6 Mutants.” *The Journal of Comparative Neurology* 510 (1): 79–99.

Zhou, Jian, and Olga G. Troyanskaya. 2015. “Predicting Effects of Noncoding Variants with Deep Learning-Based Sequence Model.” *Nature Methods* 12 (10): 931–34.